# Explaining Hyperparameter Optimization
# via Partial Dependence Plots

**Julia Moosbauer**\*, **Julia Herbinger**\*, **Giuseppe Casalicchio, Marius Lindauer, Bernd Bischl**
Department of Statistics, Ludwig-Maximilians-University Munich, Munich, Germany
Institute of Information Processing, Leibniz University Hannover, Hannover, Germany
`{julia.moosbauer, julia.herbinger, giuseppe.casalicchio,`
`bernd.bischl}@stat.uni-muenchen.de`
`lindauer@tnt.uni-hannover.de`

## Abstract

Automated hyperparameter optimization (HPO) can support practitioners to obtain peak performance in machine learning models. However, there is often a lack of valuable insights into the effects of different hyperparameters on the final model performance. This lack of explainability makes it difficult to trust and understand the automated HPO process and its results. We suggest using interpretable machine learning (IML) to gain insights from the experimental data obtained during HPO with Bayesian optimization (BO). BO tends to focus on promising regions with potential high-performance configurations and thus induces a sampling bias. Hence, many IML techniques, such as the *partial dependence plot* (PDP), carry the risk of generating biased interpretations. By leveraging the posterior uncertainty of the BO surrogate model, we introduce a variant of the PDP with estimated confidence bands. We propose to partition the hyperparameter space to obtain more confident and reliable PDPs in relevant sub-regions. In an experimental study, we provide quantitative evidence for the increased quality of the PDPs within sub-regions.

## 1 Introduction

Most machine learning (ML) algorithms are highly configurable. Their hyperparameters must be chosen carefully, as their choice often impacts the model performance. Even for experts, it can be challenging to find well-performing hyperparameter configurations. Automated machine learning (AutoML) systems and methods for automated HPO have been shown to yield considerable efficiency compared to manual tuning by human experts [Snoek et al., 2012]. However, these approaches mainly return a well-performing configuration and leave users without insights into decisions of the optimization process. Questions about the importance of hyperparameters or their effects on the resulting performance often remain unanswered. Not all data scientists trust the outcome of an AutoML system due to the lack of transparency [Drozdal et al., 2020]. Consequently, they might not deploy an AutoML model, despite all performance gains. Providing insights into the search process may help increase trust and facilitate interactive and exploratory processes: A data scientist could monitor the AutoML process and make changes to it (e.g., restricting or expanding the search space) already *during* optimization to anticipate unintended results.

Transparency, trust, and understanding of the inner workings of an AutoML system can be increased by interpreting the internal surrogate model of an AutoML approach. For example, BO trains a surrogate model to approximate the relationship between hyperparameter configurations and model performance. It is used to guide the optimization process towards the most promising regions of the hyperparameter space. Hence, surrogate models implicitly contain information about the influence of

---

\*These authors contributed equally to this work.

35th Conference on Neural Information Processing Systems (NeurIPS 2021).

hyperparameters. If the interpretation of the surrogate matches with a data scientist's expectation, confidence in the correct functioning of the system may be increased. If these do not match, it provides an opportunity to look either for bugs in the code or for new theoretical insights.

We propose to analyze surrogate models with methods from IML to provide insights into the results of HPO. In the context of BO, typical choices for surrogate models are flexible, probabilistic black-box models, such as Gaussian processes (GP) or random forests. Interpreting the effect of single hyperparameters on the performance of the model to be tuned is analogous to interpreting the feature effect of the black-box surrogate model. We focus on the PDP [Friedman, 2001], which is a widely-used method[2] to visualize the average marginal effect of single features on a black-box model's prediction. When applied to surrogate models, they provide information on how a specific hyperparameter influences the estimated model performance. However, applying PDPs out of the box to surrogate models might lead to misleading conclusions. Efficient optimizers such as BO tend to focus on exploiting promising regions of the hyperparameter space while leaving other regions less explored. Therefore, a sampling bias in input space is introduced, which in turn can lead to a poor fit and biased interpretations in underexplored regions of the space.

**Contributions:** We study the problem of sampling bias in experimental data produced by AutoML systems and the resulting bias of the surrogate model and assess its implications on PDPs. We then derive an uncertainty measure for PDPs of probabilistic surrogate models. In addition, we propose a method that splits the hyperparameter space into interpretable sub-regions of varying uncertainty to obtain sub-regions with more reliable and confident PDP estimates. In the context of BO, we provide evidence for the usefulness of our proposed methods on a synthetic function and in an experimental study in which we optimize the architecture and hyperparameters of a deep neural network. Our Supplementary Material provides (A) more background related to uncertainty estimates, (B) notes on how our methods are applied to hierarchical hyperparameter spaces, (C) details on the experimental setup and more detailed results, (D) a link to the source code.

**Reproducibility and Open Science**: The implementation of the proposed methods as well as reproducible scripts for the experimental analysis are provided in a public git-repository[3].

## 2   Background and Related Work

Recent research has begun to question whether the evaluation of an AutoML system should be purely based on the generated models' predictive performance without considering interpretability [Hutter et al., 2014a, Pfisterer et al., 2019, Freitas, 2019, Xanthopoulos et al., 2020]. Interpreting AutoML systems can be categorized as (1) interpreting the resulting ML model on the underlying dataset, or (2) interpreting the HPO process itself. In this paper, we focus on the latter.

Let $c : \Lambda \to \mathbb{R}$ be a *black-box* cost function, mapping a hyperparameter configuration $\boldsymbol{\lambda} = (\lambda_1, ..., \lambda_d)$ to the model error[4] obtained by a learning algorithm with configuration $\boldsymbol{\lambda}$. The hyperparameter space may be mixed, containing categorical and continuous hyperparameters. The goal of HPO is to find $\boldsymbol{\lambda}^* \in \arg\min_{\boldsymbol{\lambda} \in \Lambda} c(\boldsymbol{\lambda})$. Throughout the paper, we assume that a surrogate model $\hat{c} : \Lambda \to \mathbb{R}$ is given as an approximation to $c$. If the surrogate is assumed to be a GP, $\hat{c}(\boldsymbol{\lambda})$ is a random variable following a Gaussian posterior distribution. In particular, for any finite indexed family of hyperparameter configurations $(\boldsymbol{\lambda}^{(1)}, ..., \boldsymbol{\lambda}^{(k)}) \in \Lambda^k$, the vector of estimated performance values is Gaussian with a posterior mean $\hat{\boldsymbol{m}} = (\hat{m}(\boldsymbol{\lambda}^{(i)}))_{i=1,...,k}$ and covariance $\hat{\boldsymbol{K}} = (\hat{k}(\boldsymbol{\lambda}^{(i)}, \boldsymbol{\lambda}^{(j)}))_{i,j=1,...,k}$.

**Hyperparameter Importance.** Understanding which hyperparameters influence model performance can provide valuable insights into the tuning strategy [Probst et al., 2019]. To quantify relevance of hyperparameters, models that inherently quantify feature relevance – e.g., GPs with ARD kernel [Neil, 1996] – can be used as surrogate models. Hutter et al. [2014a] quantified the importance of hyperparameters based on a random forest fitted on data generated by BO, for which the importance of both the main and the interaction effects of hyperparameters was calculated by a functional ANOVA approach. Similarly, Sharma et al. [2019] quantified the hyperparameter importance of

---

[2]There exist various implementations [Greenwell, 2017, Pedregosa et al., 2011]), extensions [Greenwell et al., 2018, Goldstein et al., 2015] and applications [Friedman and Meulman, 2003, Cutler et al., 2007].

[3]`https://github.com/slds-lmu/paper_2021_xautoml`

[4]Typically, the model error is estimated via cross-validation or hold-out testing.

residual neural networks. These works highlight how useful it is to quantify the importance of hyperparameters. However, importance scores do not show *how* a specific hyperparameter affects the model performance according to the surrogate model. Therefore, we propose to visualize the assumed marginal effect of a hyperparameter. A model-agnostic interpretation method that can be used for this purpose is the PDP.

**PDPs for Hyperparameters.** Let $S \subset \{1, 2, ..., d\}$ denote an index set of features, and let $C = \{1, 2, ..., d\} \setminus S$ be its complement. The partial dependence (PD) function [Friedman, 2001] of $c : \Lambda \to \mathbb{R}$ for hyperparameter(s) $S$ is defined as[5]

$$c_S(\boldsymbol{\lambda}_S) := \mathbb{E}_{\boldsymbol{\lambda}_C}[c(\boldsymbol{\lambda})] = \int_{\Lambda_C} c(\boldsymbol{\lambda}_S, \boldsymbol{\lambda}_C) \, \mathrm{d}\mathbb{P}(\boldsymbol{\lambda}_C). \tag{1}$$

When analyzing the PDP of hyperparameters, we are usually interested in how their values $\boldsymbol{\lambda}_S$ impact model performance uniformly across the hyperparameter space. In line with prior work [Hutter et al., 2014a], we therefore assume $\mathbb{P}$ to be the uniform distribution over $\Lambda_C$. Computing $c_S(\boldsymbol{\lambda}_S)$ exactly is usually not possible because $c$ is unknown and expensive to evaluate in the context of HPO. Thus, the posterior mean $\hat{m}$ of the probabilistic surrogate model $\hat{c}(\boldsymbol{\lambda})$ is commonly used as a proxy for $c$. Furthermore, the integral may not be analytically tractable for arbitrary surrogate models $\hat{c}$. Hence, the integral is approximated by Monte Carlo integration, i.e.,

$$\hat{c}_S(\boldsymbol{\lambda}_S) \quad = \quad \frac{1}{n} \sum\nolimits_{i=1}^{n} \hat{m}\left(\boldsymbol{\lambda}_S, \boldsymbol{\lambda}_C^{(i)}\right) \tag{2}$$

for a sample $\left(\boldsymbol{\lambda}_C^{(i)}\right)_{i=1,...,n} \sim \mathbb{P}(\boldsymbol{\lambda}_C)$. $\hat{m}\left(\boldsymbol{\lambda}_S, \boldsymbol{\lambda}_C^{(i)}\right)$ represents the marginal effect of $\boldsymbol{\lambda}_S$ for one specific instance $i$. Individual conditional expectation (ICE) curves [Goldstein et al., 2015] visualize the marginal effect of the $i$-th observation by plotting the value of $\hat{m}\left(\boldsymbol{\lambda}_S, \boldsymbol{\lambda}_C^{(i)}\right)$ against $\boldsymbol{\lambda}_S$ for a set of grid points[6] $\boldsymbol{\lambda}_S^{(g)} \in \Lambda_S, g \in \{1, ..., G\}$. Analogously, the PDP visualizes $\hat{c}_S(\boldsymbol{\lambda}_S)$ against the grid points. Following from Eq. 2, the PDP visualizes the average over all ICE curves. In HPO, the marginal predicted performance is a related concept. Instead of approximating the integral via Monte Carlo, the integral over $\hat{c}$ is computed exactly. Hutter et al. [2014a] propose an efficient approach to compute this integral for random forest surrogate models.

**Uncertainty Quantification in PDPs.** Quantifying the uncertainty of PDPs provides additional information about the reliability of the mean estimator. Hutter et al. [2014a] quantified the model uncertainty specifically for random forests as surrogates in BO by calculating the standard deviation of the marginal predictions of the individual trees. However, this procedure is not applicable to general probabilistic surrogate models, such as the commonly used GP. There are approaches that quantify the uncertainty for ML models that do not provide uncertainty estimates out-of-the-box. Cafri and Bailey [2016] suggested a bootstrap approach for tree ensembles to quantify the uncertainties of effects based on PDPs. Another approach to quantify the uncertainty of PDPs is to leverage the ICE curves. For example, Greenwell [2017] implemented a method that marginalizes over the mean ± standard deviation of the ICE curves for each grid point. However, this approach quantifies the underlying uncertainty of the data at hand rather than the model uncertainty, as explained in Appendix A.1. A model-agnostic estimate based on uncertainty estimates for probabilistic models is missing so far.

**Subgroup PDPs.** Recently, a new research direction concentrates on finding more reliable PDP estimates within subgroups of observations. Molnar et al. [2020] focused on problems in PDP estimation with correlated features. To that end, they apply transformation trees to find homogeneous subgroups and then visualize a PDP for each subgroup. Grömping [2020] looked at the same problem and also uses subgroup PDPs, where ICE curves are grouped regarding a correlated feature. Britton [2019] applied a clustering approach to group ICE curves to find interactions between features. However, none of these approaches aim at finding subgroups where reliable PDP estimates have low uncertainty. Additionally, to the best of our knowledge, nothing similar exists for analyzing experimental data created by HPO.

---

[5]To keep notation simple, we denote $c(\boldsymbol{\lambda})$ as a function of two arguments $(\boldsymbol{\lambda}_S, \boldsymbol{\lambda}_C)$ to differentiate components in the index set $S$ from those in the complement. The integral shall be understood as a multiple integral of $c$ where $\boldsymbol{\lambda}_j, j \in C$, are integrated out.

[6]Grid points are typically chosen as an equidistant grid or sampled from $\mathbb{P}(\boldsymbol{\lambda}_S)$. The granularity $G$ is chosen by the user. For categorical features, the granularity typically corresponds to the number of categories.

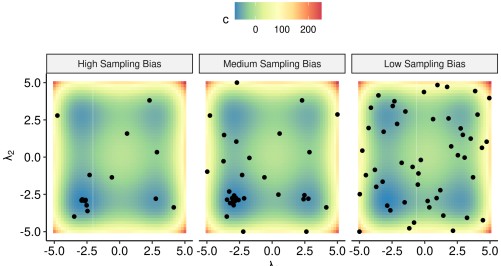

Figure 1: Illustration of the sampling bias when optimizing the $2D$ Styblinski Tang function with BO and the Lower Confidence Bound (LCB) acquisition function $a(\boldsymbol{\lambda}) = \hat{m}(\boldsymbol{\lambda}) + \tau \cdot \hat{s}(\boldsymbol{\lambda})$ for $\tau = 0.1$ (left) and $\tau = 2$ (middle) vs. data sampled uniformly at random (right).

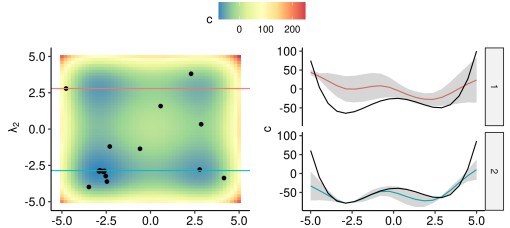

Figure 2: The two horizontal cuts (left) yield two ICE curves (right) showing the mean prediction and uncertainty band against $\lambda_1$ for $\hat{c}$ with $\tau = 0.1$ on the $2D$ Styblinski-Tang function. The upper ICE curve deviates more from the true effect (black) and shows a higher uncertainty.

## 3 Biased Sampling in HPO

Visualizing the marginal effect of hyperparameters of surrogate models via PDPs can be misleading. We show that this problem is due to the sequential nature of BO, which generates dependent instances (i.e., hyperparameter configurations) and thereby introduces a sampling and a resulting model bias. To save computational resources in contrast to grid search or random search, efficient optimizers like BO tend to exploit promising regions of the hyperparameter space while other regions are less explored (see Figure 1). Consequently, predictions of surrogate models are usually more accurate with less uncertainty in well-explored regions and less accurate with high uncertainty in under-explored regions. This model bias also affects the PD estimate (see Figure 2). ICE curves may be biased and less confident if they are computed in poorly-learned regions where the model has not seen much data before. Under the assumption of uniformly distributed hyperparameters, poorly-learned regions are incorporated in the PD estimate with the same weight as well-learned regions. ICE curves belonging to regions with high uncertainty may obfuscate well-learned effects of ICE curves belonging to other regions when they are aggregated to a PDP. Hence, the model bias may also lead to a less reliable PD estimate. PDPs visualizing only the mean estimator of Eq. (2) do not provide insights into the reliability of the PD estimate and how it is affected by the described model bias.

## 4 Quantifying Uncertainty in PDPs

Pointwise uncertainty estimates of a probabilistic model provide insights into the reliability of the prediction $\hat{c}(\boldsymbol{\lambda})$ for a specific configuration $\boldsymbol{\lambda}$. This uncertainty directly correlates with how explored the region around $\boldsymbol{\lambda}$ is. Hence, including the model's uncertainty structure into the PD estimate enables users to understand in which regions the PDP is more reliable and which parts of the PDP must be cautiously interpreted.[7] We now extend the PDP of Eq. (2) to probabilistic surrogate models $\hat{c}$ (e.g., a GP). Let $\boldsymbol{\lambda}_S$ be a fixed grid point and $\left(\boldsymbol{\lambda}_C^{(i)}\right)_{i=1,\ldots,n} \sim \mathbb{P}\left(\boldsymbol{\lambda}_C\right)$ a sample that is used to compute the Monte

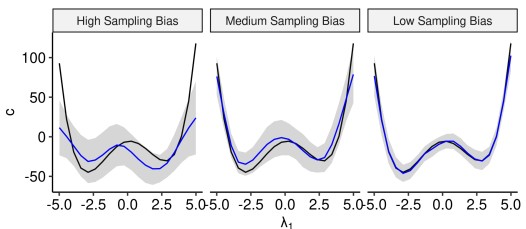

Figure 3: PDPs (blue) with confidence bands for surrogates trained on data created by BO and LCB with $\tau = 0.1$ (left), $\tau = 1$ (middle) and uniform i.i.d. dataset (right) vs. the true PD (black).

Carlo estimate of Eq. (2). The vector of predicted performances at the grid point $\boldsymbol{\lambda}_S$ is $\hat{\boldsymbol{c}}(\boldsymbol{\lambda}_S) = \left(\hat{c}\left(\boldsymbol{\lambda}_S, \boldsymbol{\lambda}_C^{(i)}\right)\right)_{i=1,\ldots,n}$ with (posterior) mean $\hat{\boldsymbol{m}}(\boldsymbol{\lambda}_S) := \left(\hat{m}\left(\boldsymbol{\lambda}_S, \boldsymbol{\lambda}_C^{(i)}\right)\right)_{i=1,\ldots,n}$ and

---

[7]Note that we aim at representing model uncertainty in a PD estimate, and not the variability of the mean prediction (see Appendix A.1 for a more detailed justification).

a (posterior) covariance $\hat{K}\left(\boldsymbol{\lambda}_S\right) := \left(\hat{k}\left(\left(\boldsymbol{\lambda}_S, \boldsymbol{\lambda}_C^{(i)}\right), \left(\boldsymbol{\lambda}_S, \boldsymbol{\lambda}_C^{(j)}\right)\right)\right)_{i,j=1,...,n}$. Thus, $\hat{c}_S\left(\boldsymbol{\lambda}_S\right) = \frac{1}{n}\sum_{i=1}^n \hat{c}\left(\boldsymbol{\lambda}_S, \boldsymbol{\lambda}_C^{(i)}\right)$ is a random variable itself. The expected value of $\hat{c}_S\left(\boldsymbol{\lambda}_S\right)$ corresponds to the PD of the posterior mean function $\hat{m}$ at $\boldsymbol{\lambda}_S$, i.e.:

$$\hat{m}_S\left(\boldsymbol{\lambda}_S\right) \;=\; \mathbb{E}_{\hat{c}}\left[\hat{c}_S\left(\boldsymbol{\lambda}_S\right)\right] = \mathbb{E}_{\hat{c}}\left[\frac{1}{n}\sum_{i=1}^n \hat{c}\left(\boldsymbol{\lambda}_S, \boldsymbol{\lambda}_C^{(i)}\right)\right] = \frac{1}{n}\sum_{i=1}^n \hat{m}\left(\boldsymbol{\lambda}_S, \boldsymbol{\lambda}_C^{(i)}\right). \quad (3)$$

The variance of $\hat{c}_S\left(\boldsymbol{\lambda}_S\right)$ is

$$\hat{s}_S^2(\boldsymbol{\lambda}_S) \;=\; \mathbb{V}_{\hat{c}}\left[\hat{c}_S\left(\boldsymbol{\lambda}_S\right)\right] = \mathbb{V}_{\hat{c}}\left[\frac{1}{n}\sum_{i=1}^n \hat{c}\left(\boldsymbol{\lambda}_S, \boldsymbol{\lambda}_C^{(i)}\right)\right] = \frac{1}{n^2}\mathbf{1}^\top \hat{K}\left(\boldsymbol{\lambda}_S\right)\mathbf{1}. \quad (4)$$

For the above estimate, it is important that the kernel is correctly specified such that the covariance structure is modeled properly by the surrogate model. Eq. (4) can be approximated empirically by treating the pairwise covariances as unknown, i.e.:

$$\hat{s}_S^2\left(\boldsymbol{\lambda}_S\right) \;\approx\; \frac{1}{n}\sum_{i=1}^n \hat{K}\left(\boldsymbol{\lambda}_S\right)_{i,i}. \quad (5)$$

In Appendix A.2, we show empirically that this approximation is less sensitive to kernel misspecifications. Please note that the variance estimate and the mean estimate can also be applied to other probabilistic models, such as GAMLSS[8], transformation trees, or a random forest. An example for PDPs with uncertainty estimates is shown in Figure 3 for different degrees of a sampling bias.

## 5 Regional PDPs via Confidence Splitting

As discussed in Section 3, (efficient) optimization may imply that the sampling is biased, which in turn can produce misleading interpretations when IML is naively applied. We now aim to identify sub-regions $\Lambda' \subset \Lambda$ of the hyperparameter space in which the PD can be estimated with high confidence, and separate those from sub-regions in which it cannot be estimated reliably. In particular, we identify sub-regions in which poorly-learned effects do not obfuscate the well-learned effects along each grid point, thereby allowing the user to draw conclusions with higher confidence. By partitioning the entire hyperparameter space through a tree-based approach into disjoint and interpretable sub-regions, a more detailed understanding of the sampling process and hyperparameter effects is achieved. Users can either study the hyperparameter effect of a (confident) sub-region individually or understand the exploration-exploitation sampling of HPO by considering the complete tree structure. The result of this procedure for a single split is shown in Figure 5.

The PD estimate on the *entire* hyperparameter space $\Lambda$ is computed by sampling the Monte Carlo estimate $(\boldsymbol{\lambda}_C^{(i)})_{i\in\mathcal{N}} \sim \mathbb{P}(\boldsymbol{\lambda}_C)$, $\mathcal{N} := \{1, 2, ..., n\}$. We now introduce the PD estimate on a *sub-region* $\Lambda' \subset \Lambda$ simply as $(\boldsymbol{\lambda}_C^{(i)})_{i\in\mathcal{N}'}$ only using $\mathcal{N}' = \{i \in \mathcal{N}\}_{\boldsymbol{\lambda}^{(i)} \in \Lambda'}$. Since we are interested in the marginal effect of the hyperparameter(s) $S$ at each $\boldsymbol{\lambda}_S \in \Lambda_S$, we will usually visualize the PD for the whole range $\Lambda_S$. Thus, all obtained sub-regions should be of the form $\Lambda' = \Lambda_S \times \Lambda'_C$ with $\Lambda'_C \subset \Lambda_C$. This corresponds to an average of ICE curves in the set $i \in \mathcal{N}'$. The pseudo-code to partition a hyperparameter (sub-)space $\Lambda$ and corresponding sample $(\boldsymbol{\lambda}_C^{(i)})_{i\in\mathcal{N}} \in \Lambda_C$, $\mathcal{N} \subseteq \{1, ..., n\}$, into two child regions is shown in Algorithm 1. This splitting is recursively applied in a CART[9]-like procedure [Breiman et al., 1984b] to expand a full tree structure, with the usual stopping criteria (e.g., a maximum number of splits, a minimum size of a region, or a minimum improvement in each node). In each leaf node, the sub-regional PDP and its corresponding uncertainty estimate are computed by aggregating over all contained ICE curves.

The criterion to evaluate a specific partitioning is based on the idea of grouping ICE curves with similar uncertainty structure. To be more exact, we evaluate the impurity of a PD estimate on a sub-region $\Lambda'$ with the help of the associated set of observations $\mathcal{N}' = \{i \in \mathcal{N}\}_{\boldsymbol{\lambda}_C^{(i)} \in \Lambda'_C}$, also referred to as nodes, as follows: For each grid point $\boldsymbol{\lambda}_S$, we use the L2 loss in $L\left(\boldsymbol{\lambda}_S, \mathcal{N}'\right)$ to evaluate how the

---

[8]Generalized additive models for location, scale and shape
[9]Classification and regression trees

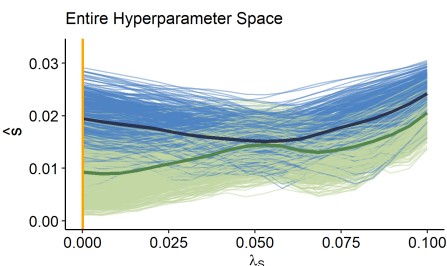

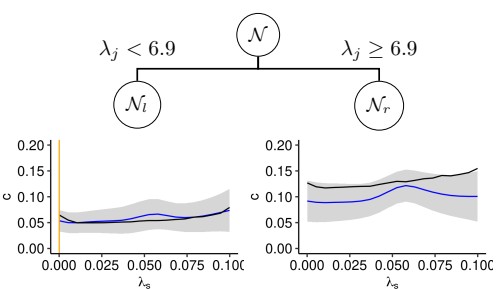

Figure 4: ICE curves of $\hat{s}$ of $\boldsymbol{\lambda}_S$ for the left (green) and right (blue) sub-region after the first split. The darker lines represent the respective PDPs. The orange vertical line marks the value $\lambda_S$ of the optimal configuration.

Figure 5: Example of two estimated PDPs (blue line) and $95\%$ confidence bands after one partitioning step. The orange vertical line is the value of $\boldsymbol{\lambda}_S$ from the optimal configuration, the black curve is the true PD estimate $c_S(\boldsymbol{\lambda}_S)$.

uncertainty varies across all ICE estimates $i \in \mathcal{N}'$ using $\hat{s}^2_{S|\mathcal{N}'}(\boldsymbol{\lambda}_S) := \frac{1}{|\mathcal{N}'|}\sum_{i\in\mathcal{N}'}\hat{s}^2\left(\boldsymbol{\lambda}_S, \boldsymbol{\lambda}_C^{(i)}\right)$ and aggregate the loss $\mathcal{L}(\boldsymbol{\lambda}_S, \mathcal{N}')$ over all grid points in $\mathcal{R}_{L2}(\mathcal{N}')$:

$$\mathcal{L}(\boldsymbol{\lambda}_S, \mathcal{N}') = \sum_{i\in\mathcal{N}'}\left(\hat{s}^2\left(\boldsymbol{\lambda}_S, \boldsymbol{\lambda}_C^{(i)}\right) - \hat{s}^2_{S|\mathcal{N}'}(\boldsymbol{\lambda}_S)\right)^2 \text{ and } \mathcal{R}_{L2}(\mathcal{N}') = \sum_{g=1}^{G}\mathcal{L}(\boldsymbol{\lambda}_S^{(g)}, \mathcal{N}'). \quad (6)$$

---

**Algorithm 1:** Tree-based Partitioning

**input:** $\mathcal{N}$
**for** $j \in C$ **do**
  **for** Every split $t$ on hyperparameter $\lambda_j$ **do**
    $\mathcal{N}_l^{j,t} = \{i \in \mathcal{N}\}_{\lambda_j^{(i)} \leq t}$
    $\mathcal{N}_r^{j,t} = \{i \in \mathcal{N}\}_{\lambda_j^{(i)} > t}$
    $\mathcal{I}(j,t) = \mathcal{R}_{L_2}(\mathcal{N}_l^{j,t}) + \mathcal{R}_{L_2}(\mathcal{N}_r^{j,t})$
  **end for**
**end for**
Choose $\left(j^*, t^*_{\lambda_j^*}\right) \in \arg\min_{j,t}\mathcal{I}(j,t)$
Return $\mathcal{N}_l^{j,t}$ and $\mathcal{N}_r^{j,t}$ for $(j,t) = \left(j^*, t^*_{\lambda_j^*}\right)$

---

Hence, we measure the pointwise $L_2$-distance between ICE curves of the variance function $\hat{s}^2(\boldsymbol{\lambda}_S, \boldsymbol{\lambda}_C^{(i)})$ and its PD estimate $\hat{s}^2_{S|\mathcal{N}'}(\boldsymbol{\lambda}_S)$ within a sub-region $\mathcal{N}'$. This seems reasonable, as ICE curves in well-explored regions of the search space should, on average, have a lower uncertainty than those in less-explored regions. However, since we only split according to hyperparameters in $C$ but not in $S$, the partitioning does not cut off less explored regions w.r.t. $\boldsymbol{\lambda}_S$. Thus, the chosen split criterion groups ICE curves of the uncertainty estimate such that we receive sub-regions associated with low costs $c$ (and thus high relevance for a user) to be more confident in well-explored regions of $\boldsymbol{\lambda}_S$ and less confident in under-explored regions. Figure 4 shows that ICE curves of the uncertainty measure with high uncertainty over the entire range of $\boldsymbol{\lambda}_S$ are grouped together (right sub-region). Those with low uncertainty close to the optimal configuration of $\boldsymbol{\lambda}_S$ and increasing uncertainties for less suitable configurations are grouped together by curve similarities in the left sub-region. The respective PDPs are illustrated in Figure 5, where the confidence band in the left sub-region decreased compared to the confidence band of the global PDP especially for grid points close to the optimal value of $\boldsymbol{\lambda}_S$. Hence, by grouping observations with similar ICE curves of the variance function, resulting sub-regional PDPs with confidence bands provide the user with the information of which sub-regions of $\Lambda_C$ are well-explored and lead to more reliable PDP estimates. Furthermore, the user will know which ranges of $\boldsymbol{\lambda}_S$ can be interpreted reliably and which ones need to be regarded with caution.

To sum up, the splitting procedure provides interpretable, disjoint sub-regions of the hyperparameter space. Based on the defined impurity measure, PDPs with high reliability can be identified and analyzed. In particular, the method provides more confident and reliable estimates in the sub-region containing the optimal configuration. Which PDPs are most interesting to explore depends on the question the user would like to answer. If the main interest lies in understanding the optimization and exploring the sampling process, a user might want to keep the number of sub-regions relatively low by performing only a few partitioning steps. Subsequently, one would investigate the overall structure of the sub-regions and the individual sub-regional PDPs. If users are more interested in interpreting

hyperparameter effects only in the most relevant sub-region(s), they may want to split deeper and only look at sub-regions that are more confident than the global PDP.

Due to the nature of the splitting procedure, the PDP estimate on the entire hyperparameter space is a weighted average of the respective sub-regional PDPs. Hence, the global PDP estimate is decomposed into several sub-regional PDP estimates. Furthermore, note that the proposed method does not assume a numeric hyperparameter space, since the uncertainty estimates as well as ICE and PDP estimates can also be calculated for categorical features. Thus, it is applicable to problems with mixed spaces as long as a probabilistic surrogate model – and particularly its uncertainty estimates – are available. In Appendix B we describe how our method is applied to hierarchical hyperparameter spaces.

Since the proposed method is an instance of the CART algorithm, finding the optimal split for a categorical variable with $q$ levels generally involves checking $2^q$ subsets. This becomes computationally infeasible for high values of $q$. It remains an open question for future work if this can be sped by an optimal procedure as in regression with L2 loss [Fisher, 1958] and binary classification [Breiman et al., 1984a] or by a clever heuristic as for multiclass classification Wright and König [2019].

# 6 Experimental Analysis

In this section, we validate the effectiveness of the introduced methods. We formulate two main hypotheses: First, experimental data affected by the sampling bias lead to biased surrogate models and thus to unreliable and misleading PDPs. Second, the proposed partitioning allows us to identify an interpretable sub-region (around the optimal configuration) that yields a more reliable and confident PDP estimate. In a first experiment, we apply our methods to BO runs on a synthetic function. In this controlled setup, we investigate the validity of our hypotheses with regards to problems of different dimensionality and different degrees of sampling bias. In a second experiment, we evaluate our PDP partitioning in the context of HPO for neural networks on a variety of tabular datasets.

We assess the sampling bias of the optimization design points by comparing their empirical distribution to a uniform distribution via Maximum Mean Discrepancy (MMD) [Gretton et al., 2012, Molnar et al., 2020], which is covered in more detail in the Appendix C.1. We measure the reliability of a PDP, i.e., the degree to which a user can rely on the estimate of the PD estimate, by comparing it to the true PD $c_S(\boldsymbol{\lambda}_S)$ as defined in Eq. (1). More specifically, for every grid point $\boldsymbol{\lambda}_S^{(g)}$, we compute the negative log-likelihood (NLL) of $c_S(\boldsymbol{\lambda}_S)$ under the distribution of $\hat{c}_S(\boldsymbol{\lambda}_S)$ pointwise for every grid point $\boldsymbol{\lambda}_S^{(g)}$. The confidence of a PDP is illustrated by the width of its confidence bands $\hat{m}_S(\boldsymbol{\lambda}_S) \pm q_{1-\alpha/2} \cdot \hat{s}_S(\boldsymbol{\lambda}_S)$, with $q_{1-\alpha/2}$ denoting the $(1-\alpha/2)$-quantile of a standard normal distribution. We measure the confidence by assessing $\hat{s}_S(\boldsymbol{\lambda}_S)$ pointwise for every grid point. In particular, we consider the mean confidence (MC) across all grid points $\frac{1}{G}\sum_{g=1}^{G}\hat{s}\left(\boldsymbol{\lambda}_S^{(g)}\right)$ as well as the confidence at the grid point closest to $\hat{\boldsymbol{\lambda}}_S$ abbreviated by OC, with $\hat{\boldsymbol{\lambda}}$ being the best configuration evaluated by the optimizer. To evaluate the performance of the confidence splitting, we report the above metrics on the sub-region that contains the best configuration evaluated by the optimizer, assuming that this region is of particular interest for a user of HPO. PDPs are computed with regards to single features for $G = 20$ equidistant grid points and $n = 1000$ Monte Carlo samples.

## 6.1 BO on a Synthetic Function

We consider the $d$-dimensional Styblinski-Tang function $c : [-5,5]^d \rightarrow \mathbb{R}$, $\boldsymbol{\lambda} \mapsto \frac{1}{2}\sum_{i=1}^{d}\left(\boldsymbol{\lambda}_i^4 + 16\boldsymbol{\lambda}_i^2 + 5\boldsymbol{\lambda}_i\right)$ for $d \in \{3,5,8\}$. Since the PD is the same for each dimension $i$, we only present the effects of $\boldsymbol{\lambda}_1$. We performed BO with a GP surrogate model with a Matérn-3/2 kernel and the LCB acquisition function $a(\boldsymbol{\lambda}) = \hat{m}(\boldsymbol{\lambda}) + \tau \cdot \hat{s}(\boldsymbol{\lambda})$ with different values $\tau \in \{0.1, 1, 5\}$ to control the sampling bias. We compute the global PDP with confidence bands estimated according to Eq. (5) for the GP surrogate model $\hat{c}$ that was fitted in the *last* iteration of BO. We ran Algorithm 1, and computed the PDP in the sub-region containing the optimal configuration. All computations were repeated 30 times. Further details on the setup are given in Appendix C.2.1.

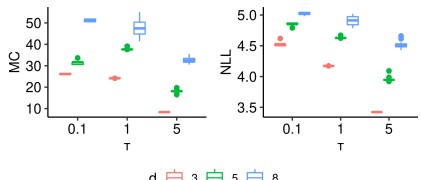

Figure 6: The figure presents the MC (left) and the NLL (right) for $d \in \{3, 5, 8\}$ for a high ($\tau = 0.1$), medium ($\tau = 1$), and low ($\tau = 5$) sampling bias across 30 replications. With a lower sampling bias, we obtain narrower confidence bands and a lower NLL.

Table 1: The table shows the relative improvement of the MC and the NLL via Algorithm 1 with 1 and 3 splits, compared to the global PDP along with the sampling bias for a $\tau = 0.1$ (high), $\tau = 2$ (medium), and $\tau = 5$ (low). Results are averaged across 30 replications.

| | | $\delta$ MC (%) | | $\delta$ NLL (%) | |
|---|---|---|---|---|---|
| $d$ | MMD | $n_{sp} = 1$ | $n_{sp} = 3$ | $n_{sp} = 1$ | $n_{sp} = 3$ |
| 3 | low (0.18) | 7.65 | 13.64 | 5.89 | 10.92 |
| 3 | medium (0.51) | 12.86 | 36.92 | 4.78 | 7.70 |
| 3 | high (0.56) | 16.52 | 34.84 | 2.77 | -1.62 |
| 5 | low (0.15) | 6.63 | 15.45 | 2.82 | 6.05 |
| 5 | medium (0.45) | 19.67 | 37.28 | 4.05 | 7.80 |
| 5 | high (0.53) | 11.99 | 33.06 | -3.86 | -1.93 |
| 8 | low (0.11) | 3.58 | 9.67 | 0.84 | 2.40 |
| 8 | medium (0.42) | 8.86 | 23.03 | 1.51 | 3.30 |
| 8 | high (0.56) | 6.59 | 19.84 | 1.53 | 4.29 |

As presented in Figure 6, the PDPs for surrogate models trained on *less biased* data (measured by the MMD) yield *lower* values of the NLL, as well as *lower* values for the MC. Table 1 shows that a single tree-based split reduces the MC by up to almost 20%, and up to 37% when performing 3 partitioning steps. Additionally, the NLL improves with an increasing number of partitioning steps in most cases. The results on the synthetic functions support our second hypothesis that the tree-based partitioning improves the reliability in terms of the NLL and the confidence of the PD estimates. The improvement of the MC is higher for a medium to high sampling bias, compared to scenarios that are less affected by sampling bias. We observe that (particularly for high sampling bias) there are some outlier cases in which the NLL worsens. More detailed results are shown in Appendix C.3.1.

## 6.2 HPO on Deep Learning

In a second experiment, we investigate HPO in the context of a surrogate benchmark [Eggensperger et al., 2015] based on the LCBench data [Zimmer et al., 2021]. For each of the 35 different OpenML [Vanschoren et al., 2013] classification tasks, LCBench provides access to evaluations of a deep neural network on 2000 configurations randomly drawn from the configuration space defined by Auto-PyTorch Tabular (see Table 5 in Appendix C.2). For each task, we trained a random forest as an empirical performance model that predicts the balanced validation error of the neural network for a given configuration. These empirical performance models serve as cheap to evaluate objective functions, which efficiently approximate the result of the real-world experiment of running a deep learning configuration on an LCBench instance. BO then acts on this empirical performance model as its objective[10].

For each task, we ran BO to obtain the optimal architecture and hyperparameter configuration. Again, we used a GP with a Matérn-3/2 kernel and LCB with $\tau = 1$. Each BO run was allotted a budget of 200 objective function evaluations. We computed the PDPs and their confidences, which are estimated according to Eq. (5), based on the surrogate model $\hat{c}$ after the final iteration. We performed tree-based partitioning with up to 6 splits based on a uniformly distributed dataset of size $n = 1000$. All computations were statistically repeated 30 times. Further details are provided in Appendix C.2.2.

For the real-world data example, we focus on answering the second hypothesis, i.e., whether the tree-based Algorithm 1 improves the reliability of the PD estimates. We compare the PDP in sub-regions after 6 splits with the global PDP. We computed the relative improvement of the confidence (MC and OC) and the NLL of the sub-regional PDP compared to the respective estimates for the global PDP. As shown in Table 2, the MC of the PDPs is on average reduced by 30% to 52%, depending on the hyperparameter. At the optimal configuration $\hat{\boldsymbol{\lambda}}_S$, the improvement even increases to $50\% - 62\%$. Thus, PDP estimates for all hyperparameters are on average – independent of the underlying dataset – clearly more confident in the relevant sub-regions when compared to the global PD estimates, especially around the optimal configuration $\hat{\boldsymbol{\lambda}}_S$. In addition to the MC, the NLL simultaneously improves. In Appendix C.3.2, we provide details regarding the evaluated metrics on the level of the dataset and demonstrate that our split criterion outperforms other impurity measures regarding MC

---

[10]Please note that the random forest is only used as a surrogate in order to construct an efficient benchmark objective, and not as a surrogate in the BO algorithm, where we use a GP.

and OC. Furthermore, we emphasize in Appendix C.3.2 the significance of our results by providing a comparison to a naive baseline method.

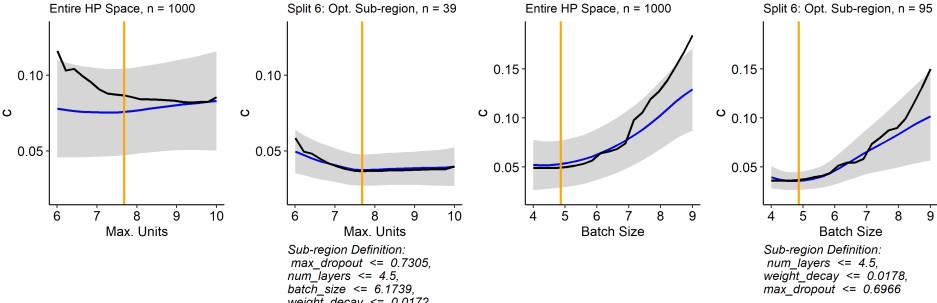

Figure 7: PDP (blue) and confidence band (grey) of the GP for hyperparameter *max. number of units* (*batch size*) on the left (right) side. The black line shows the PDP of the meta surrogate model representing the true PDP estimate. The orange vertical line marks the optimal configuration $\hat{\lambda}_S$. The relative improvements from the global PDP to the sub-regional PDP after 6 splits are for *max. number of units* (*batch size*): $\delta$ MC = 61.6% (28.4%), $\delta$ OC = 63.5% (62.2%), $\delta$ NLL = 48.6% (30.1%).

To further study our suggested method, we now highlight a few individual experiments. We chose one iteration of the *shuttle* dataset. On the two left plots of Figure 7, we see that the true PDP estimate for *max. number of units* is decreasing, while the globally estimated PDP trend is increasing and thus misleading. Although the confidence band already indicates that the PDP cannot be reliably interpreted on the entire hyperparameter space, it remains challenging to draw any conclusions from it. After performing 6 splits, we receive a confident and reliable PD estimate on an interpretable sub-region. The same plots are depicted for the hyperparameter *batch size* on the right part of Figure 7. This

Table 2: Relative improvement of MC, OC, and NLL on hyperparameter level. The table shows the respective mean (standard deviation) of the average relative improvement over 30 replications for each dataset and 6 splits.

| Hyperparameter | $\delta$ MC (%) | $\delta$ OC (%) | $\delta$ NLL (%) |
|---|---|---|---|
| Batch size | 40.8 (14.9) | 61.9 (13.5) | 19.8 (19.5) |
| Learning rate | 50.2 (13.7) | 57.6 (14.4) | 17.9 (20.5) |
| Max. dropout | 49.7 (15.4) | 62.4 (11.9) | 17.4 (18.2) |
| Max. units | 51.1 (15.2) | 58.6 (12.7) | 24.6 (22.0) |
| Momentum | 51.7 (14.5) | 58.3 (12.7) | 19.7 (21.7) |
| Number of layers | 30.6 (16.4) | 50.9 (16.6) | 13.8 (32.5) |
| Weight decay | 36.3 (22.6) | 61.0 (13.1) | 11.9 (19.7) |

example illustrates that the confidence band might not always shrink uniformly over the entire range of $\lambda_S$ during the partitioning, but often particularly around the optimal configuration $\hat{\lambda}_S$.

## 7    Discussion and Conclusion

In this paper, we showed that partial dependence estimates for surrogate models fitted on experimental data generated by efficient hyperparameter optimization can be unreliable due to an underlying sampling bias. We extended PDPs by an uncertainty estimate to provide users with more information regarding the reliability of the mean estimator. Furthermore, we introduced a tree-based partitioning approach for PDPs, where we leverage the uncertainty estimator to decompose the hyperparameter space into interpretable, disjoint sub-regions. We showed with two experimental studies that we generate, on average, more confident and more reliable regional PDP estimates in the sub-region containing the optimal configuration compared to the global PDP.

One of the main limitations of PDPs is that they bear the risk of providing misleading results if applied to correlated data in the presence of interactions, especially for nonparametric models [Grömping, 2020]. However, existing alternatives that visualize the global marginal effect of a feature such as accumulated local effect (ALE) plots [Apley and Zhu, 2020] do also not provide a fully satisfying solution to this problem [Grömping, 2020]. As a solution to this problem, Grömping [2020] suggests stratified PDPs by conditioning on a correlated and potentially interacting feature to group ICE curves. This idea is in the spirit of our introduced tree-based partitioning algorithm. However, in the context of BO we might assume the distribution in Eq. (1) to be uniform and therefore no correlations are present. Instead of correlated features, we are faced with a sampling bias (see Section 3) where

we observe regions of varying uncertainty. Hence, instead of stratifying with respect to correlated features and aggregating ICE curves in regions with less correlated features, we stratify with respect to uncertainty and aggregate ICE curves in regions with low uncertainty variation. Nonetheless, it might be interesting to compare our approach with approaches based on the considerations made by Grömping [2020] – or potentially improved ALE curves.

Another limitation when using single-feature PDPs as in our examples is that hyperparameter interactions are not visible. However, two-way interactions can be visualized by plotting two-dimensional PDPs within sub-regions. Another possibility to detect interactions is to look at ICE curves within the sub-regions. If the shape of ICE curves within a sub-region is very heterogeneous, it indicates that the hyperparameter under consideration interacts with one of the other hyperparameters. Hence, having the additional possibility to look at ICE curves of individual observations within a sub-region is an advantage compared to other global feature effect plots such as ALE plots [Apley and Zhu, 2020], as they are not defined on an observational level. While we mainly discussed GP surrogate models on a numerical hyperparameter space in our examples, our methods are applicable to a wide variety of distributional regression models and also for mixed and hierarchical hyperparameter spaces. We also considered in Appendix C.3.2 different impurity measures. While the one introduced in this paper performed best in our experimental settings, this impurity measure as well as other components are exchangeable within the proposed algorithm. In the future, we will study our method on more complex, hierarchical configuration spaces for neural architecture search.

The proposed interpretation method is based on a surrogate and consequently does provide insights about what the AutoML system has *learned*, which in turn allows plausibility checks and may increase trust in the system. To what extent this allows conclusions on the *true* underlying hyperparameter effects depends on the quality of the surrogate. How to efficiently perform model diagnostics to ensure a high surrogate quality before applying interpretability techniques is subject to future research.

While we focused on providing better explanations without generating any additional experimental data, it might be interesting to investigate in future work how confidence and reliability of IML methods can be increased most efficiently when a user is allowed to conduct additional experiments.

Overall, we believe that increasing interpretability of AutoML will pave the way for human-centered AutoML. Our vision is that users will be able to better understand the reasoning and the sampling process of AutoML systems and thus can either trust and accept the results of the AutoML system or interact with it in a feedback loop based on the gained insights and their preferences. How users can then best interact with AutoML (beyond simple changes of the configuration space) will be left open for future research.

## Acknowledgments and Disclosure of Funding

This work has been partially supported by the German Federal Ministry of Education and Research (BMBF) under Grant No. 01IS18036A. The authors of this work take full responsibilities for its content.

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
