# A  Uncertainty Estimation

## A.1  Choice of Uncertainty Quantification

Besides using the uncertainty estimate of the surrogate model to quantify the uncertainty for the PDP mean estimate (our method), it is also possible to estimate uncertainty w.r.t. the variance over different ICE curves. However, if the uncertainty was estimated via computing the variance over ICE curves, we describe how the *levels of the mean prediction vary* along the $\boldsymbol{\lambda}_S$ dimensions. In contrast, we propose to capture *model uncertainty* along the $\boldsymbol{\lambda}_S$ dimensions. For example, consider a constant surrogate function $\hat{c}(\boldsymbol{\lambda}) = \gamma$ with high uncertainty estimation $\hat{s}^2(\boldsymbol{\lambda}) = 100$. Computing the variance over ICE curves on this example will result in an uncertainty estimate of $0$ (all ICE curves are identical). Our method, however, would return a variance estimate of $100$ and thus capture model uncertainty.

## A.2  Covariance Estimates under Misspecification of Kernels

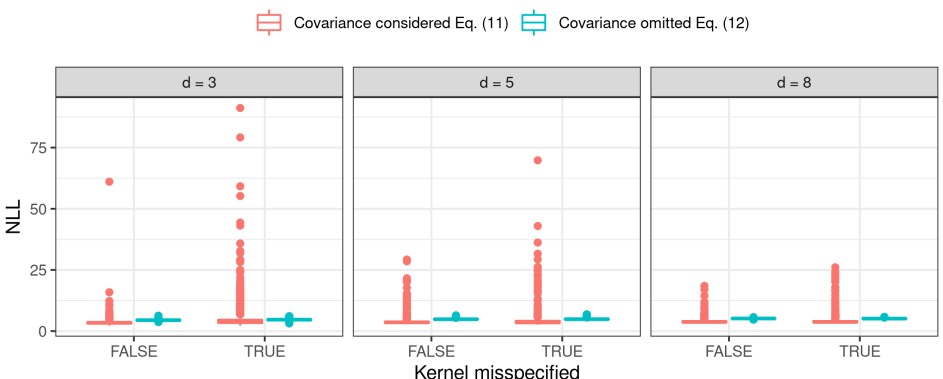

Figure 8: The figures show the NLL of the true PDP $c_1(\boldsymbol{\lambda}_1)$ under the estimated PDPs with variance estimates (4) and (5) and for a misspecified kernel (Gaussian) and a correctly specified kernel (Matérn-3/2), respectively.

Table 3: The table shows the NLL of the true PDP $c_1(\boldsymbol{\lambda}_1)$ under the estimated PDPs with variance estimates (4) and (5) and for a misspecified kernel (Gaussian) and a correctly specified kernel (Matérn-3/2), respectively. Shown are the mean across $50$ replications, and the standard deviation in brackets.

|   | Correct specification | | Misspecification | |
| --- | --- | --- | --- | --- |
| d | Estimate (5) | Estimate (4) | Estimate (5) | Estimate (4) |
| 3 | 3.61 (2.02) | 4.47 (0.27) | 5.10 (5.91) | 4.62 (0.32) |
| 5 | 3.93 (2.00) | 4.87 (0.23) | 4.33 (3.72) | 4.89 (0.28) |
| 8 | 4.05 (1.12) | 5.18 (0.14) | 4.24 (2.12) | 5.13 (0.17) |

In order to provide evidence for the claim that estimate Eq. (4) is more sensitive to misspecifications in the kernel (and thus in the covariance structure) than Eq. (5), we performed some prior experiments.

We assume that we are given an objective function that is generated by a Gaussian process (GP) with a Matérn-3/2 kernel. In our experiments, that function was created by fitting a GP on tuples $\left(\boldsymbol{\lambda}^{(i)}, y^{(i)}\right)_{i=1,\ldots,30}$, with $\boldsymbol{\lambda}^{(i)} \sim \mathrm{Unif}\left([-5,5]^d\right)$ and $y^{(i)}$ corresponding to the value of the $d$-dimensional Styblinski Tang function for $\boldsymbol{\lambda}^{(i)}$. The posterior mean of this GP will further serve as our true objective $c$ to pretend that we know the correct kernel specification of the ground-truth. Subsequently, we fit both a GP surrogate model with correctly specified kernel (i.e., a Matérn-3/2 kernel) and a surrogate model with a misspecified kernel (in our case, we chose a Gaussian kernel) to the data $\left(\boldsymbol{\lambda}^{(i)}, c\left(\boldsymbol{\lambda}^{(i)}\right)\right)_{i=1,\ldots,30}$. In both cases, we compute the PDPs for $\boldsymbol{\lambda}_1$ with both variance

estimates Eq. (4) and Eq. (5) and measure the negative log-likelihood (NLL) of $c_S$ under the respective estimated PDP. We performed 50 repetitions of the experiments for $d \in \{3, 5, 8\}$, respectively.

Figure 8 shows that the median of the NLL across all 50 replications is *slightly* lower for the covariance estimate in Eq. (4). However, the variance of the NLL is much higher for estimate in Eq. (4) as compared to Eq. (5). Table 3 confirms that, when using variance estimate Eq. (4), the standard deviation of the NLL values is lower. We conclude that the reliability of the estimate is particularly sensitive to a correct choice of the kernel function. The NLL for the PDPs computed with variance estimate Eq. (5) is - independent of whether the kernel is correctly specified or not - less sensitive to misspecifications in the kernel.

# B    Hierarchical Hyperparameter Spaces

Search spaces in HPO and AutoML are often hierarchical, i.e., some hyperparameters are only active conditional on the value of another hyperparameter (the latter usually being a categorical choice, e.g., using a certain method). The underlying dependency structure can be visualized by a tree structure (Figure 9 (a)). If one hyperparameter can activate another hyperparameter, we call the former a *parent* and the latter is *subordinate* to the former and called a *child*. A hyperparameter that has no parents is called a global hyperparameter. Sampled configurations can be presented in a nested block matrix (Figure 9 (b)), with missing entries for inactive hyperparameters.

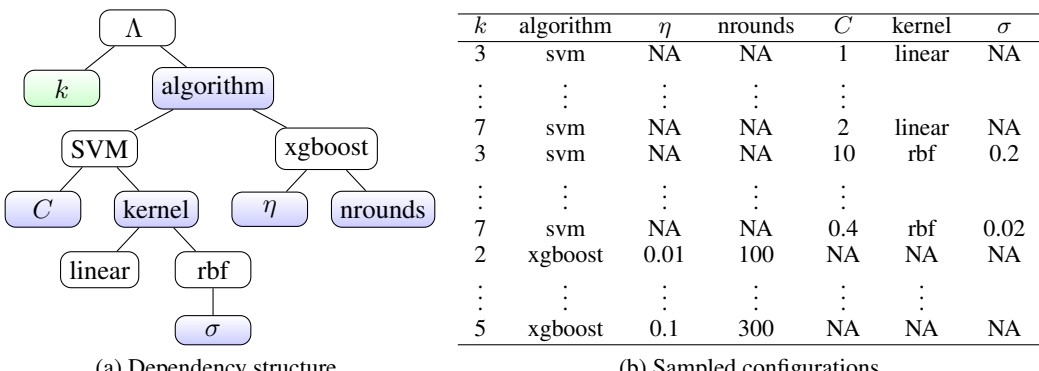

| $k$ | algorithm | $\eta$ | nrounds | $C$ | kernel | $\sigma$ |
|---|---|---|---|---|---|---|
| 3 | svm | NA | NA | 1 | linear | NA |
| ⋮ | ⋮ | ⋮ | ⋮ | ⋮ | | |
| 7 | svm | NA | NA | 2 | linear | NA |
| 3 | svm | NA | NA | 10 | rbf | 0.2 |
| ⋮ | ⋮ | ⋮ | ⋮ | ⋮ | | |
| 7 | svm | NA | NA | 0.4 | rbf | 0.02 |
| 2 | xgboost | 0.01 | 100 | NA | NA | NA |
| ⋮ | ⋮ | ⋮ | ⋮ | ⋮ | ⋮ | |
| 5 | xgboost | 0.1 | 300 | NA | NA | NA |

(a) Dependency structure         (b) Sampled configurations

Figure 9: Hyperparameter $k \in \mathbb{N}$ represents the number of selected components of PCA, applied as a preprocessing step. While it is global, always active, and has no subordinate children, *algorithm* – dependent on its values *SVM* and *xgboost* – is parent to hyperparameters $C$, *kernel*, $\eta$ and *nrounds*.

Various surrogate models exist, which can learn on such hierarchical spaces, e.g., GPs with specialized kernels [Levesque et al., 2017, Swersky et al., 2014], specialized trees in random forests [Hutter et al., 2014b] or handling the missing entries through imputation [Bischl et al., 2018].

The specialized tree algorithm (in normal AutoML, without PDP) can be briefly summarized as follows: We run a normal recursive partitioning as in CART, but in each split, a hyperparameter is only then eligible for potential splitting if the path leading to the current node satisfies all of its preconditions and therefore activates it.

In order to generalize this tree building in the context of hierarchical dependency structures to a regional PDP computation as in Section 5, we now make the following three modifications:

First, after having selected a hyperparameter $\boldsymbol{\lambda}_S$ for which we want to estimate the PDP, we subset the uniformly sampled test data on which we are going to fit our regional PDP algorithm to all rows for which $\boldsymbol{\lambda}_S$ is not missing, i.e., only to configurations in which $\boldsymbol{\lambda}_S$ is active[11].

Second, we now run the specialized tree algorithm as described above, but never allow to split on $\boldsymbol{\lambda}_S$. Obviously, this will then never activate any child of $\boldsymbol{\lambda}_S$, so we can never split on these.

---

[11] This will likely result in constant values for hyperparameters in the path leading to $\boldsymbol{\lambda}_S$, and consequently the tree will not split on these.

Third, we now adapt the estimation of the PDPs in a given tree node and the associated data block $\mathcal{N}$

$$\hat{c}_S(\boldsymbol{\lambda}_S) = \frac{1}{|\mathcal{N}|} \sum_{i \in \mathcal{N}} \hat{m}\left(\boldsymbol{\lambda}_S, \boldsymbol{\lambda}_C^{(i)}\right) \qquad \hat{s}_S^2(\boldsymbol{\lambda}_S) = \frac{1}{|\mathcal{N}|} \sum_{i \in \mathcal{N}} \hat{s}^2\left(\boldsymbol{\lambda}_S, \boldsymbol{\lambda}_C^{(i)}\right) \qquad (7)$$

in the presence of hierarchical structures. If hyperparameter $\boldsymbol{\lambda}_S$ is a parent, sampling from the marginal $\boldsymbol{\lambda}_C \sim \mathbb{P}(\boldsymbol{\lambda}_C)$ (which refers to all hyperparameters except $\boldsymbol{\lambda}_S$) can still yield invalid combinations $(\boldsymbol{\lambda}_S, \boldsymbol{\lambda}_C)$.

We now simply average only over valid configurations w.r.t. to the dependency structure. Formally, let $q(\boldsymbol{\lambda}) : \Lambda \longrightarrow \{0, 1\}$ be a binary predicate function which is 1 if and only if $\boldsymbol{\lambda}$ is valid w.r.t. to the dependency structure. Now let $v(\boldsymbol{\lambda}_S, \mathcal{N}) = \{i \in \mathcal{N} | q(\boldsymbol{\lambda}_S, \boldsymbol{\lambda}_C^{(i)}) = 1\} \subset \mathcal{N}$ be the set of valid configurations in $\mathcal{N}$ w.r.t. the dependency structure if we insert a given $\boldsymbol{\lambda}_S$ value. The dependency adapted PDPs now are:

$$\hat{c}_S(\boldsymbol{\lambda}_S) = \frac{1}{|v(\boldsymbol{\lambda}_S, \mathcal{N})|} \sum_{i \in v(\boldsymbol{\lambda}_S, \mathcal{N})} \hat{m}\left(\boldsymbol{\lambda}_S, \boldsymbol{\lambda}_C^{(i)}\right) \qquad (8)$$

with an analogous modification for $\hat{s}_S^2(\boldsymbol{\lambda}_S)$.

For example, to calculate the PDP of a child hyperparameter such as `nrounds` w.r.t. the example in Figure 9, we first need to subset the test dataset to all rows for which `nrounds` is not missing (in this specific example this is the same as only keeping the instances where `algorithm` takes the value `xgboost`). Due to the dependency structure only $k$, `algorithm` and $\eta$ are active hyperparameters in $\boldsymbol{\lambda}_C$. To calculate the PDP of a parent hyperparameter such as `algorithm`, there are (in this example) no missings w.r.t. $\boldsymbol{\lambda}_S$. However, we will obtain invalid configurations, e.g., when we replace the value `svm` by `xgboost` for the parent hyperparameter `algorithm`. Thus, we need to use the third modification from the above described adjustments and average only over all valid configurations w.r.t. to the dependency structure: if we insert `svm`, then the invalid configurations are dropped and we average only over those configurations that were activated when choosing `svm`, i.e., that contain non missing values in the child hyperparameters of the `svm` algorithm. If we insert `xgboost` then we only average over those configurations that contain non missing values for $\eta$ and `nrounds`.

## C  Experimental Analysis

### C.1  Maximum Mean Discrepancy

In Section 6.1 we analyze the first hypothesis how the sampling bias affects the PDP estimation. An indicator of the size of the sampling bias is the exploration factor $\tau$. The smaller $\tau$ the higher the sampling bias compared to a uniformly distributed dataset (e.g. see Figure 1). To put it in other words, the sampling bias can be quantified by the distributional shift between a uniformly distributed sample and the sample generated by the BO process. A commonly used measure to quantify such distributional differences is the Kullback-Leibler divergence. However, since the joint distribution of the generated sample is not known, the Kullback-Leibler divergence might not be the most appropriate measure here. Another metric that tests if two different samples belong to the same distribution, is the maximum mean discrepancy (MMD) [Gretton et al., 2012]. It is defined by the maximum deviation in expectation and based on the function class of reproducing kernel Hilbert space (RKHS). This is equivalent to 'the norm of the difference between distribution feature means in the RKHS' [Gretton et al., 2012].

An unbiased empirical estimate of the MMD for samples $X = \{\boldsymbol{x}^{(1)}, ..., \boldsymbol{x}^{(n)}\}$ and $Y = \{\boldsymbol{y}^{(1)}, ..., \boldsymbol{y}^{(m)}\}$ is given by

$$\begin{aligned}
\text{MMD}^2(X, Y) = {} & \frac{1}{n(n-1)} \sum_{i=1}^{n} \sum_{j \neq i}^{n} k\left(\boldsymbol{x}^{(i)}, \boldsymbol{x}^{(j)}\right) + \frac{1}{m(m-1)} \sum_{i=1}^{m} \sum_{j \neq i}^{m} k\left(\boldsymbol{y}^{(i)}, \boldsymbol{y}^{(j)}\right) \\
& - \frac{2}{nm} \sum_{i=1}^{n} \sum_{j=1}^{m} k\left(\boldsymbol{x}^{(i)}, \boldsymbol{y}^{(j)}\right)
\end{aligned}$$

It follows that for $X$ and $Y$ being drawn from the same distribution, the MMD is small while it becomes large for increasing distributional differences.

Here, $X$ represents the sample that is drawn from a uniform distribution over the hyperparameter space $\Lambda$, while $Y$ is the sample generated by the BO process. The kernel $k$ is chosen to be the radial basis function kernel with parameter $\sigma$ being set to the median L2-distance between sample points. This heuristic is commonly used [Gretton et al., 2012].

## C.2 Experimental Design

All experiments only require CPUs (and no GPUs) and were computed on a Linux cluster (see Table 4).

Table 4: Description of the infrastructure used for the experiments in this paper.

| Computing Infrastructure | |
| --- | --- |
| Type | Linux CPU Cluster |
| Architecture | 28-way Haswell-EP nodes |
| Cores per Node | 1 |
| Memory limit (per core) | 2.2 GB |

The computational complexity of the PDP estimation with uncertainty is $\mathcal{O}(G \cdot n) \cdot \mathcal{O}(\hat{c})$, with $\mathcal{O}(\hat{c})$ being the runtime complexity of single surrogate prediction, $n$ denoting the size of the dataset to compute the Monte Carlo estimate and $G$ being the number of grid points. In the context of HPO, the general assumption is that the evaluation time of $\hat{c}$ is negligibly low as compared to evaluation $c$. So we argue that the runtime complexity of computing a PDP with uncertainty estimate can be neglected in this context. When computing ICE curves and their variance estimates beforehand, the algorithmic complexity of Algorithm 1 corresponds to the algorithmic complexity of the tree splitting [Breiman et al., 1984b].

In our experiments, the runtimes to compute the PDPs and perform the tree splitting lies within a few minutes. We consider them to be negligible and will thus not report these.

### C.2.1 BO on a Synthetic Function

The Styblinski-Tang function

$$c : [-5, 5]^d \quad \rightarrow \quad \mathbb{R} \tag{9}$$

$$\boldsymbol{\lambda} \quad \mapsto \quad \frac{1}{2} \sum_{i=1}^{d} \left( \boldsymbol{\lambda}_i^4 + 16 \boldsymbol{\lambda}_i^2 + 5 \boldsymbol{\lambda}_i \right) \tag{10}$$

was optimized via BO for $d \in \{3, 5, 8\}$ with a total budget of $\{80, 150, 250\}$ objective function evaluations, respectively, to allow sufficient optimization progress depending on the problem dimension.

We computed an initial random design of size $4d$[12]. We performed BO with a GP surrogate model with a Matérn-3/2 kernel and the LCB acquisition function $a(\boldsymbol{\lambda}) = \hat{m}(\boldsymbol{\lambda}) + \tau \cdot \hat{s}(\boldsymbol{\lambda})$ with different values $\tau \in \{0.1, 1, 5\}$. A nugget $10^{-8}$ was added for for numerical stability. We denote the best evaluated configuration, measured by $\hat{c}$, by $\hat{\boldsymbol{\lambda}}$.

Based on the last surrogate model, we performed the partitioning in Algorithm 1 for a total number of 5 splits, with the different splitting criteria (see Section C.3.2), with PDPs being computed with a $G = 20$ equidistant grid points, and $n = 1000$ samples for the Monte Carlo approximation[13].

For all subsequent analysis, we considered the subregions $\Lambda'$, for which $\hat{\boldsymbol{\lambda}} \in \Lambda'$, and computed the PDPs according to Estimate (5).

---

[12]The initial design was fixed across replications

[13]Both grid-points and the data to compute the MC estimate are fixed across replications

Table 5: Hyperparameter space 1 of Auto-PyTorch Tabular.

| Name | Range | log | type |
|------|-------|-----|------|
| Number of layers | $[1, 5]$ | no | int |
| Max. number of units | $[64, 512]$ | yes | int |
| Batch size | $[16, 512]$ | yes | int |
| Learning rate (SGD) | $[1e^{-4}, 1e^{-1}]$ | yes | float |
| Weight decay | $[1e^{-5}, 1e^{-1}]$ | no | float |
| Momentum | $[0.1, 0.99]$ | no | float |
| Max. dropout rate | $[0.0, 1.0]$ | no | float |

Table 6: Hyperparameter space of the random forest that was tuned over to compute the empirical performance model.

| Name | Range | log | type |
|------|-------|-----|------|
| Number of trees | $[10, 500]$ | yes | int |
| mtry | $\{true, false\}$ | no | bool |
| Minimum Size of Nodes | $[1, 5]$ | no | int |
| Number of Random Splits | $[1, 100]$ | no | int |

For every subregions considered $\Lambda'$, we compute a partial dependence of $c$ for feature $\boldsymbol{\lambda}_1$, denoted as $c_1(\boldsymbol{\lambda}_1)$ to establish a ground-truth PDP estimate.

### C.2.2 MLP

All experimental data were downloaded from the LCBench project[14]. As an empirical performance model, we fitted a random forest (ranger) to approximate the relationship between hyperparameters and balanced error rate (BER). For every dataset, we performed a random search with $500$ iterations and evaluation via 3-fold cross-validation to choose reasonable for the hyperparameters represented in Table 6. The empirical performance model acts as ground-truth in our experiments, and thus, we denote it by $c$. This function was used to compute the true PDP $c_S$.

We computed an initial random design of size $2 \cdot d$[15]. We performed BO with a GP surrogate model with a Matérn-3/2 kernel and the LCB acquisition function $a(\boldsymbol{\lambda}) = \hat{m}(\boldsymbol{\lambda}) + \tau \cdot \hat{s}(\boldsymbol{\lambda})$ with $\tau = 1$. A nugget effect was modeled. The maximum budget per BO run was set to $200$ objective function evaluations. We denote the best evaluated configuration, measured by $\hat{c}$, by $\hat{\boldsymbol{\lambda}}$.

Based on the last surrogate model, we performed the partitioning in Algorithm 1 for a total number of 6 splits, with the different splitting criteria (see Section C.3.2), with PDPs being computed with a $G = 20$ equidistant grid points, and $n = 1000$ samples for the Monte Carlo approximation[16].

### C.3 Detailed Results

### C.3.1 Synthetic

In Section 6.1 we analyzed our tree-based partitioning method on the Styblinski-Tang function for different dimensions and degrees of sampling bias. To make the results of Table 1 more tangible, we visualized the associated PDPs and confidence bands for $\lambda_1$ and $\tau = 1$ of one iteration in Figure 10. The plots show clearly, that the number of splits required to obtain more confident and reliable PDP estimate in the sub-region containing the optimal configuration depends on the problem dimension.

### C.3.2 MLP

In Section 6.2 we evaluated the reliability of PDP estimation for the partitioning procedure proposed in Section 5. The results presented in Section 6.2 are aggregated over a total number of 35 different

---

[14]https://github.com/automl/LCBench, Apache License 2.0

[15]The initial design was fixed across replications

[16]The grid and the data used to compute the Monte Carlo estimate was fixed across replications

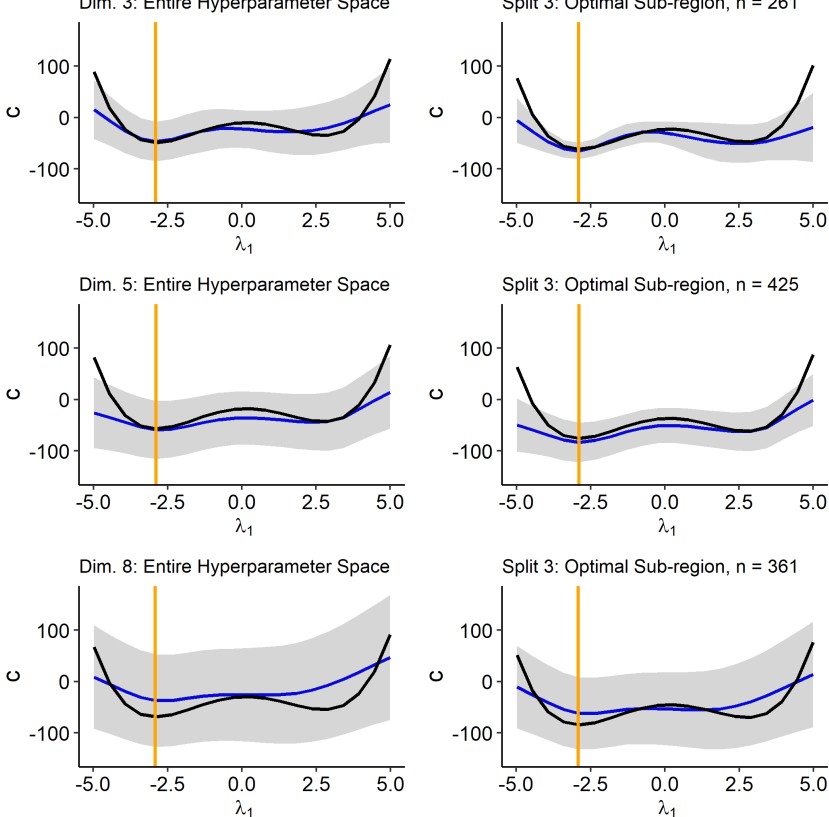

Figure 10: PDP (blue) and confidence band (grey) of the GP for hyperparameter $\lambda_1$ for the Styblinski-Tang function in case of 3 (top), 5 (middle) and 8 (bottom) dimensions. The black line shows the true PDP estimate of the Styblinski-Tang function. The orange vertical line marks the optimal configuration $\hat{\boldsymbol{\lambda}}_1$.

datasets. In Tables 7 and 8 the relative improvement of the mean confidence (MC) and NLL are presented on dataset level. The mean and standard deviation are averaged over all hyperparameters. Furthermore, the mean values of the features providing the highest and lowest relative improvement for each dataset are reported. Following on that, Table 9 shows for each hyperparameter the number of datasets for which the respective hyperparameter led to the highest (lowest) relative improvement for both evaluation metrics.

**Split Criteria** In Section 5, we introduced Eq. 6 as split criteria within the tree-based partitioning of Algorithm 1. This measure is based on splitting ICE curves based on curve similarities, which is especially suitable in the underlying context as explained in Section 5. However, we also compared it to two other measures that are based on uncertainty estimates provided by the probabilistic surrogate model. The first one is also based on ICE curves of the variance function $\hat{s}^2(\boldsymbol{\lambda}_S, \boldsymbol{\lambda}_C^{(i)})$ and its PD estimate $\hat{s}^2_{S|\mathcal{N}'}(\boldsymbol{\lambda}_S)$ within a sub-region $\mathcal{N}'$. However, instead of minimizing the distance between curves and group the associated ICE curves regarding similar behavior, we can also minimize the area under ICE curves of the variance function. The reasoning for this is as follows: If we aim for tight confidence bands over the entire range of $\Lambda_S$, we want the ICE curves of the variance function to be - on average - as low as possible. This is equivalent to minimizing the average area under ICE curves of the variance function. Thus, the calculation of Eq. 6 changes such that we first calculate the average area between each ICE curve of the uncertainty function and the respective sub-regional PDP

$$L\left(\boldsymbol{\lambda}_S, i\right) = \frac{1}{G}\sum\nolimits_{g=1}^{G}\left(\hat{s}^2\left(\boldsymbol{\lambda}_S^{(g)}, \boldsymbol{\lambda}_C^{(i)(g)}\right) - \hat{s}^2_{S|\mathcal{N}'}\left(\boldsymbol{\lambda}_S^{(g)}\right)\right),$$

Table 7: Relative improvement of MC on dataset level. The table shows the mean ($\mu$) and standard deviation ($\sigma$) of the relative improvement (in %) over all 7 hyperparameters and 30 runs after 6 splits. Additionally the mean value of the hyperparameter with the highest ($\mu_h$) and lowest ($\mu_l$) mean improvement are shown.

| Dataset | $\mu$ | $\sigma$ | $\mu_h$ | $\mu_l$ |
|---|---|---|---|---|
| adult | 34 | 6 | 38 | 25 |
| airlines | 49 | 20 | 61 | 3 |
| albert | 57 | 26 | 78 | 14 |
| Amazon_employee_access | 58 | 17 | 69 | 21 |
| APSFailure | 46 | 17 | 60 | 22 |
| Australian | 41 | 7 | 46 | 32 |
| bank-marketing | 29 | 13 | 45 | 15 |
| blood-transfusion-service | 34 | 20 | 39 | 13 |
| car | 44 | 17 | 51 | 32 |
| christine | 47 | 14 | 54 | 19 |
| cnae-9 | 66 | 26 | 83 | 7 |
| connect-4 | 47 | 14 | 56 | 17 |
| covertype | 41 | 17 | 53 | 12 |
| credit-g | 57 | 21 | 69 | 7 |
| dionis | 49 | 21 | 63 | 5 |
| fabert | 64 | 21 | 75 | 18 |
| Fashion-MNIST | 41 | 12 | 47 | 18 |
| helena | 43 | 16 | 52 | 8 |
| higgs | 42 | 14 | 52 | 17 |
| jannis | 35 | 13 | 44 | 19 |
| jasmine | 46 | 11 | 56 | 27 |
| jungle_chess_2pcs_raw | 33 | 15 | 44 | 6 |
| kc1 | 33 | 12 | 41 | 17 |
| KDDCup09_appetency | 52 | 21 | 63 | 3 |
| kr-vs-kp | 46 | 14 | 56 | 26 |
| mfeat-factors | 56 | 16 | 70 | 29 |
| MiniBooNE | 36 | 14 | 42 | 18 |
| nomao | 30 | 6 | 34 | 22 |
| numerai28.6 | 60 | 28 | 76 | -3 |
| phoneme | 29 | 7 | 32 | 25 |
| segment | 53 | 21 | 66 | 10 |
| shuttle | 48 | 11 | 58 | 32 |
| sylvine | 37 | 6 | 42 | 29 |
| vehicle | 34 | 8 | 41 | 30 |
| volkert | 44 | 16 | 55 | 12 |

Table 8: Relative improvement of the NLL on dataset level. The table shows the mean ($\mu$) and standard deviation ($\sigma$) of the relative improvement (in %) over all 7 hyperparameters and 30 runs after 6 splits. Additionally the mean value of the feature with the highest ($\mu_h$) and lowest ($\mu_l$) mean improvement are shown.

| Dataset | $\mu$ | $\sigma$ | $\mu_h$ | $\mu_l$ |
|---|---|---|---|---|
| adult | 13 | 6 | 23 | 8 |
| airlines | 17 | 9 | 23 | 1 |
| albert | 31 | 13 | 40 | 6 |
| Amazon_employee_access | -0 | 36 | 29 | -35 |
| APSFailure | 15 | 7 | 23 | 6 |
| Australian | 12 | 14 | 23 | -4 |
| bank-marketing | 7 | 9 | 17 | -1 |
| blood-transfusion-service | 6 | 17 | 10 | -8 |
| car | 26 | 32 | 35 | 10 |
| christine | 10 | 11 | 17 | 1 |
| cnae-9 | 67 | 37 | 93 | -11 |
| connect-4 | -4 | 38 | 22 | -84 |
| covertype | 28 | 13 | 38 | 8 |
| credit-g | 41 | 24 | 81 | 2 |
| dionis | 47 | 55 | 144 | -18 |
| fabert | 37 | 17 | 54 | 8 |
| Fashion-MNIST | 15 | 11 | 28 | 2 |
| helena | -20 | 31 | -9 | -35 |
| higgs | 20 | 12 | 33 | -2 |
| jannis | 17 | 7 | 21 | 8 |
| jasmine | 6 | 14 | 24 | -11 |
| jungle_chess_2pcs_raw | 9 | 15 | 24 | -7 |
| kc1 | 11 | 10 | 17 | 4 |
| KDDCup09_appetency | 23 | 28 | 62 | -33 |
| kr-vs-kp | 9 | 35 | 43 | -17 |
| mfeat-factors | 25 | 19 | 51 | 10 |
| MiniBooNE | 9 | 14 | 17 | -8 |
| nomao | 8 | 6 | 16 | 3 |
| numerai28.6 | 17 | 9 | 23 | 4 |
| phoneme | 11 | 7 | 17 | 5 |
| segment | 22 | 57 | 41 | -31 |
| shuttle | 35 | 24 | 84 | 19 |
| sylvine | 14 | 17 | 38 | -0 |
| vehicle | 0 | 20 | 9 | -14 |
| volkert | 23 | 18 | 40 | 5 |

Table 9: Number of datasets each of the hyperparameters had the highest $\mu_h$ and lowest $\mu_l$ average relative improvement w.r.t. MC and NLL.

| | MC | | NLL | |
|---|---|---|---|---|
| Hyperparameter | # $\mu_h$ | # $\mu_l$ | # $\mu_h$ | # $\mu_l$ |
| Batch size | 1 | 3 | 3 | 4 |
| Learning rate | 6 | 2 | 6 | 3 |
| Max. dropout | 9 | 1 | 2 | 1 |
| Max. units | 4 | | 7 | |
| Momentum | 8 | | 7 | 3 |
| Number of layers | 3 | 14 | 9 | 11 |
| Weight decay | 4 | 15 | 1 | 13 |

where $\hat{s}^2_{S|\mathcal{N}'}\left(\boldsymbol{\lambda}^{(g)}_S\right) := \frac{1}{|\mathcal{N}'|}\sum_{i\in\mathcal{N}'}\hat{s}^2\left(\boldsymbol{\lambda}^{(g)}_S, \boldsymbol{\lambda}^{(i)(g)}_C\right)$, and aggregate the quadratic value of it over all observations in the respective sub-region:

$$\mathcal{R}_{area}(\mathcal{N}') = \sum_{i\in\mathcal{N}'} L(\boldsymbol{\lambda}_S, i)^2. \tag{11}$$

Second, we also used the uncertainty estimates of the probabilistic surrogate model for each observation of the test data itself to define an impurity measure. Therefore we calculated the squared deviation of each observation to the mean uncertainty within the respective node. Hence, compared

to the other two approaches, we do not group curves but the observations themselves regarding their uncertainty. We further refer to this approach as the *variance (var)* approach.

As a third measure that is not based on the uncertainty estimates, we used the MSE of the posterior mean estimate of the surrogate model as split criterion. This is the most commonly used measure for regression trees and hence a solid baseline measure.

We compared the four impurity measures for the partitioning procedure over all datasets and hyperparameters. We compare the results that we presented in Section 6.2 with the according results for the other three measures in Table 10. The impurity measure based on curve similarities that we used for our analysis (L2) outperforms the other three measures on average for all hyperparameters regarding MC and especially regarding OC. With regards to NLL there is not one measure that outperforms all others, but rather all measures perform on average over all hyperparameters equally good.

Table 10: Comparison of different impurity measures regarding the relative improvement of MC, OC and NLL on hyperparameter level. The table compares the results of Table 2 (L2) with the according results for the impurity measure based on Eq. 11 (area), the *variance* measure (var) and the *mean* measure.

| Hyperparameter | $\delta$ MC (in %) | | | | $\delta$ OC (in %) | | | | $\delta$ NLL (in %) | | | |
|---|---|---|---|---|---|---|---|---|---|---|---|---|
| | L2 | area | var | mean | L2 | area | var | mean | L2 | area | var | mean |
| Batch size | 41 | 40 | 38 | 36 | 62 | 58 | 55 | 53 | 20 | 19 | 16 | 19 |
| Learning rate | 50 | 50 | 50 | 42 | 58 | 57 | 57 | 51 | 18 | 18 | 18 | 19 |
| Max. dropout | 50 | 49 | 47 | 41 | 62 | 61 | 58 | 53 | 17 | 18 | 17 | 15 |
| Max. units | 51 | 51 | 50 | 45 | 59 | 58 | 58 | 53 | 25 | 24 | 25 | 25 |
| Momentum | 52 | 51 | 51 | 43 | 58 | 57 | 57 | 53 | 20 | 20 | 20 | 16 |
| Number of layers | 31 | 30 | 29 | 25 | 51 | 46 | 46 | 45 | 14 | 15 | 15 | 13 |
| Weight decay | 36 | 35 | 34 | 29 | 61 | 53 | 51 | 52 | 12 | 12 | 11 | 10 |

**Baseline comparison**     To emphasize the significance of our results we compare our results from Section 6.2 with the following naive baseline method: We consider the L1-neighborhood around the optimal configuration – where the GP can be assumed to be quite confident due to the focused sampling of BO – which has the same size as the sub-region found by our method. We compute the PDP on this neighborhood, and compare it to the sub-regional PDP found by our method, in the same way as in Section 6.2. We calculated the average improvement of the three evaluation metrics over all datasets and repetitions on hyperparameter level as done in Table 2 of our paper. While the mean confidence for our method improves on average by 30 - 52%, the naive baseline method improves only by 8 - 23%. Close to the optimal configuration, the mean improvement of our method is between 50-62% while the baseline method only improves by 16-42%. While the negative log-likelihood does not improve for the baseline method, it improves using our method by 12-24%. See Table 11 for more detailed results. Hence, our method results in more reliable and confident PDP estimates than this baseline method. These results justify using the more complex approach of grouping ICE curves based on similarity of their uncertainty structure to receive more reliable and confident PDP estimates in the sub-region close to the optimal configuration. Other disadvantages of the baseline method are that we need to specify the size of the neighborhood and that we only receive a rather local view around the optimal configuration. Our method on the other hand decomposes the global PDP in several distinct and interpretable sub-regions which helps the user to understand which regions of the entire hyperparameter space can be interpreted more reliably and which ones need to be regarded with caution.

**Increased confidence with more splits**     Furthermore, it needs to be noted that by using our method the mean confidence and NLL improve on average if we use six splits. However, this does not mean that they improve by design when splitting into sub-regions. As shown in Tables 7 and 8, improvements heavily depend on dataset and HP. Different factors influence the optimal number of splits, such as the sampling bias, size of the test-set, and dimensionality of the HP space. For some of our benchmarks, the best results are reached with fewer splits. One example is shown in Figure 11, where improvements in both metrics are made until Split 2 and by splitting deeper, estimates get less accurate especially when sample sizes in sub-regions become very small. Thus, the number of splits is a (useful and flexible) control parameter in our method which can be determined within a

Table 11: Relative improvement of MC, OC and NLL on hyperparameter level. The table shows for our method and the baseline method the respective mean (standard deviation) of the average relative improvement over 30 replications for each dataset and 6 splits.

| | Tree-based partitioning | | | Baseline method | | |
|---|---|---|---|---|---|---|
| Hyperparameter | $\delta$ MC (%) | $\delta$ OC (%) | $\delta$ NLL (%) | $\delta$ MC (%) | $\delta$ OC (%) | $\delta$ NLL (%) |
| Batch size | 40.8 (14.9) | 61.9 (13.5) | 19.8 (19.5) | 13.7 (12.1) | 18.9 (16.0) | 1.4 (21.6) |
| Learning rate | 50.2 (13.7) | 57.6 (14.4) | 17.9 (20.5) | 23.1 (17.7) | 27.2 (20.7) | -3.4 (27.0) |
| Max. dropout | 49.7 (15.4) | 62.4 (11.9) | 17.4 (18.2) | 21.1 (16.8) | 26.7 (16.8) | 3.3 (22.1) |
| Max. units | 51.1 (15.2) | 58.6 (12.7) | 24.6 (22.0) | 19.1 (16.5) | 22.0 (17.1) | -1.4 (19.7) |
| Momentum | 51.7 (14.5) | 58.3 (12.7) | 19.7 (21.7) | 21.9 (16.4) | 25.3 (16.9) | 2.1 (25.4) |
| Number of layers | 30.6 (16.4) | 50.9 (16.6) | 13.8 (32.5) | 8.1 (5.9) | 15.4 (12.8) | 0.9 (11.8) |
| Weight decay | 36.3 (22.6) | 61.0 (13.1) | 11.9 (19.7) | 22.6 (15.9) | 41.7 (15.8) | 2.2 (24.2) |

human-in-the-loop approach (view plots after each split and stop when results are satisfying) or by defining a quantitative measure (e.g., based on a threshold for confidence improvement).

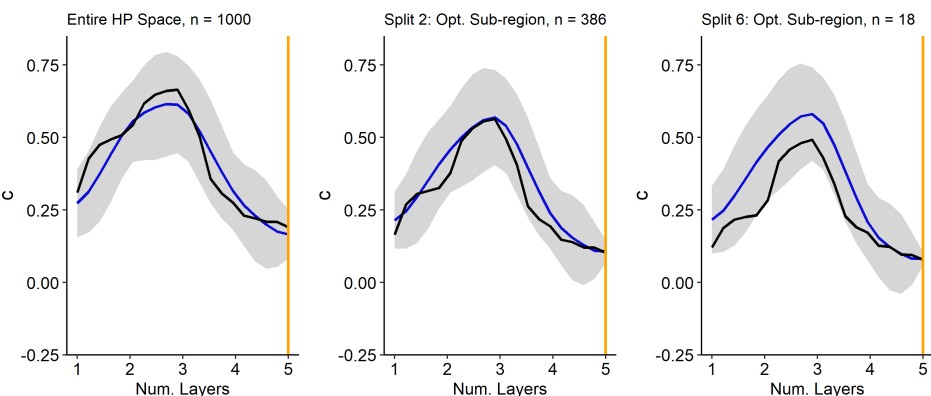

Figure 11: Estimated PDP of GP (blue) and true PDP estimate (black). The relative improvements after 2 (6) splits are $\delta$ MC = 5% (0%) and $\delta$ NLL = 5% ($-28\%$).

## D Code

All code related to this paper is made available via a public repository[17]. All methods are implemented within the folder `R`, and all code used to perform the experiments are provided in `benchmarks`. All analyses shown in this paper in form of tables or figures can be reproduced via running the notebooks in `analysis`.