# OpenReview forum: "Explaining Hyperparameter Optimization via Partial Dependence Plots"
_NeurIPS.cc/2021/Conference — NeurIPS 2021 Poster_

### Official Review · Reviewer_kioz · 2021-07-16

**Rating:** 5
**Confidence:** 3

**Summary:**

The paper presents several ideas to help understand and interpret the effects of different hyperparameters on the model chosen by automated machine learning.

The contributions include proposing an uncertainty measure for PDPs of probabilistic surrogate models including Bayesian optimization and others. To further increase interpretability, the paper suggests a procedure of dividing the hyperparameter space into subspaces/sub-regions, so that the hyperparameter effect of sub-regions can be analyzed individually.

The experiment results of the d-dimensional Styblinski-Tang function show the PDPs for surrogate models trained on less biased data produces lower values of the NLL and MC. , as well as lower values for the MC. Splitting into sub-regions helps to reduce MC generally. For high sampling bias, MC seems decrease when splitting 1 or 3 times, however, NLL can become worse in some cases.

In the other experiment, the paper studies HPO for 35 different OpenML classification tasks. The study found that the MC of the PDPs reduces significantly after 6 splits when comparing the PDP in sub-regions with the global PDP.


**Limitations And Societal Impact:**

Yes.

**Main Review:**

Using Partial Dependence Plots (PDPs) to interpret the effects of hyperparameters is not a new idea. The paper proposes a new uncertainty measure and also the idea of divide the hyperparameter space into sub-regions by constructing a tree structure. The difference with existing work is clear, and incremental, in my opinion. The paper cites many previous studies.

The paper claims the new method produces "more confident and more reliable regional PDP estimates in the sub-region containing the optimal configuration compared to the global PDP ". The experiment results confirm that claim by testing on a Synthetic Function and a set of OpenML classification tasks. The baselines in the paper use the same method proposed by the paper. It will be interesting to have results of comparing with other methods, for instance,  Greenwell [2017] or Grömping [2020].  The paper lists the limitation of the proposed approach.

The paper is well written, overall. In Algorithm 1: Tree-based Partitioning, it is not clear to me how to choose the split t. Hyperparameters can be categorical or continuous, so the optimization problem to computing t can be difficult.

I believe interpretability is an import topic for ML and AutoML. Being able to explain why a model is chosen by AutoML helps to convince people to adopt the model. The conclusion from the study is not surprising for BO based AutoML. I don't feel the proposed method produces significant results.


**Time Spent Reviewing:**

6

---

> ### Author Response · Authors · 2021-08-10
> **Response to Reviewer kioz**
>
> We thank the reviewer for the valuable review and would like to address their concerns in the following.
>
> > Comparison to a baseline
>
> We think that such a more detailed comparison has its merits. Hence, we included a comparison with a naive baseline but not with the methods proposed by Greenwell or Grömpig (as they are not appropriate here):
>
> 1. [Greenwell] Greenwell (2017) is proposing an uncertainty measure that measures the variability of ICE curves. As argued in the paper we are interested in the uncertainty of the PDP arising from uncertainty in the surrogate model, and we have shown how to derive this estimator in a principled manner from any probabilistic model. This is conceptually different from the “uncertainty estimate” proposed by Greenwell (2017) which should not be viewed as confidence bands as they measure only the variability of ICE curves, i.e. the variability of the predictions at each grid point (not the uncertainty, see Appendix A.1 for an example of this difference). It follows that the two uncertainty measures are not conceptually comparable.
> 2. [Grömping] Applying the subgrouping approach described by Grömping (2020) or Molnar (2020) will not lead to any subgroups at all in the HPO setting because they find subgroups in which features are uncorrelated. This is always the case in the HPO setting (see discussion with Reviewer APrH). To the best of our knowledge, we are the first to provide subgrouping for effect plots in the HPO setting.
> 3. However, we agree that a comparison to a more naive approach would emphasize the significance of our results. Therefore, we have added a comparison with the following naive baseline method to our experiments in Section 6.2: We consider the L1-neighborhood around the optimal configuration - where the GP can be assumed to be quite confident due to BO's focused sampling - which has the same size as the subregion found by our method. We compute the PDP on this neighborhood, and compare it to the sub-regional PDP found by our method, in the same way as in Section 6.2.
> We calculated the average improvement of the three evaluation metrics over all datasets and repetitions on hyperparameter level as done in Table 2 of our paper. While the mean confidence for our method improves on average by 30 - 52%, the naive baseline method improves only by 8 - 23%. Close to the optimal configuration, the mean improvement of our method is between 50-62% while the baseline method only improves by 16-42%. While the negative log-likelihood does not improve for the baseline method, it improves using our method by 12-24%. See https://figshare.com/s/ca1e58f619a46ed221ec for more detailed results. Hence, our method results in more reliable and confident PDP estimates than this baseline method.
>
>
> > Tree-based partitioning algorithm
>
> We agree that algorithmic details on Algorithm 1 might benefit a reader's understanding, which we will add to the additional page in case we get accepted. In general, Algorithm 1 is an instance of a general CART-like tree fitting procedure with a custom splitting criterion that encodes the target of returning reliable partial dependence plots. Thus, the question of finding a splitting point *t* generally applies to tree fitting algorithms and has already been addressed by literature [1] and efficient tree implementations.
> The computation of the optimal splitting point *t* on a continuous splitting variable is efficiently performed by ranking all *n* values of the splitting variable, so *O(n)*. In practice, this can be sped up by working on quantiles (see many tree-boosting implementations).
> For a categorical feature (with *q* levels) this is more complex, as for a naive approach each level subset must be considered - so *O(2^q)*, which works only for smaller to medium *q* - which arguably more likely in the HPO setting where we usually do not have dozens of categories. This can be sped up for certain tasks / loss-functions by using an exact presorting approach (e.g., regression with L2 loss [2], binary classification [3]),. Whether such a presorting is possible here, remains open for future research. But there are many tasks for which this presorting is als unclear (e.g. multiclass), but tree-like approaches have been applied in practice for decades on many low-to-medium-q-scenarios.
>
> [1] Wright et al., 2019, *Splitting on categorical predictors in random forests*
>
> [2] Fisher, 1958, *On grouping for maximum homogeneity*
>
> [3] Breiman, 1984, *Classification and Regression trees*
>
>
>
> > Significance of the proposed method
>
> We would like to briefly reiterate why we strongly believe that our method and the insights of our paper have major significance for the AutoML community and users of AutoML:
>
> * As we have argued in our introduction, Drosdal et al. (2020) clearly show that non-interpretability of AutoML is a major hurdle for adoption.
> * We show that a naive application of IML in the HPO setting results in wrong estimates.
> * We show substantial drops in uncertainty and reduced bias for our estimators (and Reviewer J9Xw notes this, too).
>
> If this does not address your concern: We are somewhat unsure what "unsurprising conclusion" could otherwise refer to, could you please clarify?

---

### Official Review · Reviewer_APrH · 2021-07-17

**Rating:** 6
**Confidence:** 4

**Summary:**

This paper focuses on the topic of explaining the hyperparameter optimization (HPO) process usually used in AutoML via partial dependence plots (PDP). In particular, the authors argue that naive methods suffer from sampling bias of HPO, and demonstrate how to quantify uncertainty in the resulting PDP and propose a further improvement by looking at PDP in sub-regions.

**Limitations And Societal Impact:**

1. One major disadvantage of PDP is that it can provides misleading results for correlated features, which usually happen in practice including in HPO setting. It is definitely worth some discussion in the last section.

2. The proposed method can be applied to explain ml models directly. It's worth to see how it can interpret using one or two examples.

**Main Review:**

Originality: As far as I know the method proposed is original. There are lots of work on interpretable machine learning directly applied on models, but explaining the process of hyperparameter optimization is a relatively less studies while important area.

Quality: The claims made in the paper are supported by the experiment studies. The authors are also honest about discussing both pros and cons of the proposed method.

Clarity: Overall it has good clarity. However I think the author needs more discussion in the motivation of the proposed method. See later comments. I do feel that the intro and background are a bit too long. The authors can consider moving the mathematical introduction of PDP to late sections.

Significance: I believe this topic is of importance to practitioners and data scientists.

**Time Spent Reviewing:**

3

---

> ### Author Response · Authors · 2021-08-10
> **Response to Reviewer APrH**
>
> We thank the reviewer for their feedback that our paper is original and addresses an important but understudied problem, and that our experiments support our claims. We would like to comment on the justification of PDPs as a method, and gladly discuss the reviewer’s suggestion to apply our method in the broader context of generally interpreting ML methods.
>
> > Motivation of using PDPs
>
> We agree with the reviewer that PDPs (in general!) have limitations, mainly arising from correlated input features.
> * First of all, more as a side note: An often forgotten aspect is that PDPs and correlations are only problematic if there are also interactions in the model, e.g., see Grömping (2020) or Liu, 2018, *Model interpretation: A unified derivative-based framework for nonparametric regression and supervised machine learning*. However, in many HPO applications, interactions in the surrogate are at least likely.
> * While there are alternatives to PDPs like Marginal or ALE (Apley et al., 2019, *Visualizing the Effects of Predictor Variables in Black Box Supervised Learning Models*) plots. As discussed by e.g. Friedman (2001) and Grömping (2020), those have their own drawbacks and PDPs are still SoTA to visualize effects of features.
> * PDPs have the advantage that they are based on ICE curves allowing the visualization of effects on observation level which is not possible in alternative methods such as ALE or M-Plots.
> * In contrast to the standard setting of interpreting ML models, in the HPO setting there is no natural data-distribution on the hyperparameter configuration space. Hence, we argue that estimating PDPs on an i.i.d. uniform distribution is a *natural* choice for HPO interpretation, especially because a user is interested in interpreting hyperparameter effects evenly across the space as also assumed by Hutter et al. (2014). This is the reason why we use an i.i.d. uniform distribution to calculate our new PDP estimators, where no correlations are then apparent. Hence, the main limitation of PDPs does not apply in our context.
>
> While PDPs are commonly applied by other researchers to visualize hyperparameter effects in the HPO context (see Hutter et al. (2014) and papers building upon that), we have studied the limitations of the method in this context and have identified a major limitation due to a sampling bias. We have addressed this issue by grouping ICE curves in a meaningful way (which is in the spirit of Grömping's suggestion to use stratified PDPs).
>
> We strongly agree that these are somewhat subtle but important points. We will discuss them in more detail in the paper and also think this discussion here helped to clear up the issue.
>
>
> > Extension of the method to general ML problems
>
> We agree that it would also be interesting to study the method in the more general context of interpreting feature effects of a ML model. In principle, the introduced method is very general with the only underlying requirement that the underlying ML model is probabilistic and provides pointwise uncertainty estimates. Furthermore, the PDP computation should consider the natural data generating distribution of the underlying problem in this context.
>
> To illustrate the applicability of our method, we trained a random forest on the bikesharing dataset (Fanaaee-T, 2013, *Event labeling combining ensemble detectors and background knowledge*) and have applied our method (see https://figshare.com/s/8aaf933b755d7fe60636). While the global PDP suggests that bike sharing is less popular for a high humidity (with high uncertainty as revealed by our method), after a first split we gain further insights about the peoples’ bike rental behavior depending on temperature: people seem to not care too much about humidity for low temperatures (with lower uncertainty), while they seem to care more for higher temperatures (with higher model uncertainty, though). We recommend further splits to extract more reliable effects. We will add this example as a tangible example in the Appendix.
>
> Still, we would like to emphasize that the scope of the paper addresses HPO and our method is particularly useful in contexts where the surrogate model’s quality differs largely across the hyperparameter space. An application to explain ML models directly should be studied in more detail, in future research.

---

> > ### Comment · Reviewer_APrH · 2021-08-20
> > **Thank you for your response.**
> >
> > Thank you for your detailed responses. I believe with these added contents the paper can be more comprehensive and informative.

---

> > > ### Author Response · Authors · 2021-08-20
> > > **Thanks for the positive assessment**
> > >
> > > Thank you very much for your positive assessment of our response. Of course, we will add all of these to our paper. If we get accepted, the additional page we would get will give us ample of space to definitely do this.
> > > Of course, we would be very grateful, if you could increase your rating of our paper based on your positive assessment of our response.

---

### Official Review · Reviewer_J9Xw · 2021-07-17

**Rating:** 7
**Confidence:** 4

**Summary:**

The paper suggests a modification of partial dependence plot for assessing the influence of hyperparameter on the fitting of the model in the context of Bayesian optimization. Bayesian optimization provides a surrogate function that describes the influence of hyperpatameter on the cost function with associated uncertainty. The surrogate function can be used as a proxy of the actual cost function to generate the partial dependence plot. However, this approach usually does not capture the associated uncertainty. Moreover, given Bayesian optimization explores the high-performing hyperparameter space more to find a suitable solution, it samples the hyperparameter space in a biased manner by design, and thus, it is more uncertain in region where less hyperparameters are sampled. Estimating the uncertainty of the partial dependence plot from these regions can bias the partial dependence plot. The authors resolve this issue by partitioning the hyperparameter space in regions based on the certainty of the surrogate function, and show that estimating partial dependence plot from region that has low uncertainty provides a better estimate, and the partial dependence plot becomes more interpretable.

**Limitations And Societal Impact:**

The authors are clear about the limitations of their work, i.e., proposed heuristics. The work, to my understanding, does not have any negative societal impact.

**Main Review:**

- The paper tackles an interesting problem. Explaining the contribution of hyperparamter choices in complex machine learning models helps the end-users select a hyperparameter value more informatively. Partial dependence plot is an effective way of achieving this. The proposed approach additionally provides uncertainty around the partial dependence plot, and breaks the parameter space down to estimate this uncertainty better.
- The paper is well written and explores issues with directly estimating the partial dependence plot from surrogate function, and the proposed  modifications well.
- The paper demonstrates extensively using simulated and real data that the estimated confidence interval drops by a large margin when splitting the hyperparameter space in regions over using the global estimate. This along with graphical representation of the region splits provide additional information on the effect of hyperparameter.
- The authors mention that the real-world data were modelled with deep neural network but line 290 it is mentioned that they use random forest. Table 2 and Figure 7, howeve
r, discusses deep neural network.

**Time Spent Reviewing:**

6 hours

---

> ### Author Response · Authors · 2021-08-10
> **Response to Reviewer J9Xw**
>
> Thank you for your careful (and positive) review.
>
> > The authors mention that the real-world data were modelled with deep neural networks but line 290 it is mentioned that they use random forest. Table 2 and Figure 7, however, discuss deep neural networks.
>
> The presented real-world benchmark is a surrogate benchmark derived from real data, a common practice to efficiently perform meaningful benchmarks as proposed by Eggensperger et al. (2015). It allows to speedup HPO/AutoML benchmarks and makes them more reproducible.
> This means:
>
> a) the original data is from experiments containing the performance of a deep neural network (DNN) trained for 2000 randomly sampled hyperparameter configurations (taken from Zimmer et al., 2021); configuring the DNN is the underlying task.
>
> b) a random forest (RF) model is fitted on the data in (a) and used as a surrogate objective; instead of training a DNN, we let the RF predict the performance of a DNN under a specific configuration.
>
> c) a GP model is used in (our) BO and is explained via PDPs.
> These details are covered partially in our appendix, but as this is arguably complicated, we are happy to clarify this experimental setup (Section 6.2).

---

> > ### Comment · Reviewer_J9Xw · 2021-08-23
> > **surrogate model**
> >
> > Dear authors, thanks for the clarification on RF being the surrogate model of DNN.

---

### Decision · Program_Chairs · 2021-09-27

**Decision:**

Accept (Poster)

**Comment:**

We thank the authors for the detailed clarifications they provided during the rebuttal. Model understanding is an important topic and understanding the impact/role of hyperparameters is still an open question. The reviewers all agreed that this paper makes interesting contributions even though the authors built on top of earlier work that uses partial dependency plots to interpret the role of hyperparameters. The reviewers found that the experiments supported the claims and that the discussion was balanced, covering both pros and cons of the proposed approach.